# Genome-wide analyses of non-syndromic cleft lip with palate identify 14 novel loci and genetic heterogeneity

Yanqin Yu[1,*], Xianbo Zuo[2,3,4,5,6,*], Miao He[1,7,*], Jinping Gao[2,3,4,5,6,*], Yuchuan Fu[8], Chuanqi Qin[1,8], Liuyan Meng[1], Wenjun Wang[2,3,4,5,6], Yaling Song[1], Yong Cheng[1], Fusheng Zhou[2,3,4,5,6], Gang Chen[2,3,4,5,6], Xiaodong Zheng[2,3,4,5,6], Xinhuan Wang[1], Bo Liang[2,3,4,5,6], Zhengwei Zhu[2,3,4,5,6], Xiazhou Fu[9], Yujun Sheng[2,3,4,5,6], Jiebing Hao[10], Zhongyin Liu[11], Hansong Yan[12], Elisabeth Mangold[13], Ingo Ruczinski[14], Jianjun Liu[2,3,4,5,6], Mary L. Marazita[15,16,17], Kerstin U. Ludwig[13,18], Terri H. Beaty[19], Xuejun Zhang[2,3,4,5,6,20,21], Liangdan Sun[2,3,4,5,6,22,**] & Zhuan Bian[1,**]

Non-syndromic cleft lip with palate (NSCLP) is the most serious sub-phenotype of non-syndromic orofacial clefts (NSOFC), which are the most common craniofacial birth defects in humans. Here we conduct a GWAS of NSCLP with multiple independent replications, totalling 7,404 NSOFC cases and 16,059 controls from several ethnicities, to identify new NSCLP risk loci, and explore the genetic heterogeneity between sub-phenotypes of NSOFC. We identify 41 SNPs within 26 loci that achieve genome-wide significance, 14 of which are novel (*RAD54B*, *TMEM19*, *KRT18*, *WNT9B*, *GSC/DICER1*, *PTCH1*, *RPS26*, *OFCC1/TFAP2A*, *TAF1B*, *FGF10*, *MSX1*, *LINC00640*, *FGFR1* and *SPRY1*). These 26 loci collectively account for 10.94% of the heritability for NSCLP in Chinese population. We find evidence of genetic heterogeneity between the sub-phenotypes of NSOFC and among different populations. This study substantially increases the number of genetic susceptibility loci for NSCLP and provides important insights into the genetic aetiology of this common craniofacial malformation.

[1] The State Key Laboratory Breeding Base of Basic Science of Stomatology (Hubei-MOST) and Key Laboratory of Oral Biomedicine Ministry of Education, School and Hospital of Stomatology, Wuhan University, Wuhan, Hubei 430079, China. [2] Institute of Dermatology and Department of Dermatology at No. 1 Hospital, Anhui Medical University, Hefei, Anhui 230032, China. [3] State Key Lab Incubation of Dermatology, Ministry of Science and Technology, Hefei, China. [4] Key Lab of Dermatology, Ministry of Education, Heifei, China. [5] Key Lab of Gene Resources Utilization for Severe Inherited Disorders, Anhui 230032, China. [6] Collaborative Innovation Center of Complex and Severe skin Disease, Anhui Medical University, Hefei, Anhui 230032, China. [7] Department of Pediatric Dentistry, School and Hospital of Stomatology, Wuhan University, Wuhan, Hubei 430079, China. [8] Department of Oral and Maxillofacial Surgery, School and Hospital of Stomatology, Wuhan University, Wuhan, Hubei 430079, China. [9] Department of Genetics and Centre for Developmental Biology, College of Life Science, Wuhan University, Wuhan, Hubei 430072, China. [10] The Second Charity Hospital of Henan Province, Jiaozuo, Henan 454000, China. [11] Stomatological Hospital of Nanyang, Nanyang, Henan 473013, China. [12] Stomatological Hospital of Xiangyang, Xiangyang, Hubei 441011, China. [13] Institute of Human Genetics, Life and Brain Center, University of Bonn, 53127 Bonn, Germany. [14] Department of Biostatistics, Bloomberg School of Public Health, Johns Hopkins University, Baltimore, Maryland 21205, USA. [15] Department of Oral Biology and Center for Craniofacial and Dental Genetics, School of Dental Medicine, University of Pittsburgh, Pittsburgh, Pennsylvania 15219, USA. [16] Department of Human Genetics, Graduate School of Public Health, University of Pittsburgh, Pittsburgh, Pennsylvania 15261, USA. [17] Clinical and Translational Science, Department of Psychiatry, School of Medicine, University of Pittsburgh, Pittsburgh, Pennsylvania 15213, USA. [18] Department of Genomics, Life and Brain Center, University of Bonn, 53127 Bonn, Germany. [19] Department of Epidemiology, School of Public Health, Johns Hopkins University, Baltimore, Maryland 21205, USA. [20] Department of Dermatology at No. 2 Hospital, Anhui Medical University, Hefei, Anhui 230022, China. [21] Institute of Dermatology and Department of Dermatology, Huashan Hospital of Fudan University, Shanghai 200040, China. [22] The Key Laboratory of Major Autoimmune Diseases, Anhui Province, Anhui 230032, China. * These authors contributed equally to this work. ** These authors jointly supervised this work. Correspondence and requests for materials should be addressed to L.S. (email: ahmusld@163.com) or to Z.B. (email: bianzhuan@whu.edu.cn).

O rofacial clefts (OFCs) are the most common craniofacial malformations in humans and present a major public health burden, imposing substantial health care and financial burdens on the individual, their family and society[1]. In general, the highest birth prevalence rates of OFCs are reported in Asia (especially in China and Japan), often as high as 1 in 500 and affecting more than 2.6 million people in China[2]. According to whether the patients have other malformations or anomalies, OFCs can be divided into syndromic and non-syndromic forms. Approximately 70% of cleft lip (CL) with or without cleft palate (CP) cases and 50% of CP only (CPO) cases occur, as isolated entities with no other abnormal phenotypes are considered to be non-syndromic (referred to as NSOFC)[1–3]. NSOFC is further classified into non-syndromic cleft lip with palate (NSCLP), non-syndromic CL only (NSCLO) and non-syndromic CPO (NSCPO), based on the anatomical morphology[4]. As they share common epidemiological patterns and occur during the same embryological period, NSCLP and NSCLO are often grouped together as non-syndromic CL with or without CP (NSCL/P), differing only in severity[5]. However, there is some evidence showing that NSCLP and NSCLO might harbour different genetic aetiologies[6–9].

Multiple genome-wide association study (GWAS) and relative extension studies of NSCL/P have been performed, and 22 susceptibility loci were identified[7,9–16], including the 1q32 (IRF6) locus, which was observed in previous candidate gene studies and subsequently confirmed in several GWASs[7,10,12,14–16]. However, only one GWAS of NSCL/P was conducted in a Chinese population[15] and thus the heritability in the risk of NSOFC remains unexplained in China, especially for the three distinct sub-groups of NSCLP, NSCLO and NSCPO in both Chinese and European populations.

To facilitate the understanding of the genetic architecture and gain a better understanding of the genetic basis underlying the sub-phenotypes of NSOFC, here we perform a NSCLP GWAS using two independent case–control samples from China and replicate interesting markers in a total of 23,463 samples from sub-phenotypes of NSOFC and multiple ethnic groups. We identify 14 new loci and confirm 12 previously reported ones for NSCLP. These susceptibility variants identified in the current study collectively account for 10.94% of the heritability for NSCLP in Chinese population. In addition, evidence of genetic heterogeneity is observed between the three sub-phenotypes of NSOFC and among different populations.

## Results

**Identification of 26 NSCLP-associated loci.** In the discovery stage, we genotyped 900,015 single-nucleotide polymorphisms (SNPs) using the Illumina HumanOmniZhongHua-8 BeadChip in 2,096 cases and 4,051 controls of Chinese ancestry (cohort 1). After quality control, 803,202 SNPs (call rate > 95% and minor allele frequency (MAF) > 1%) in 2,033 NSCLP cases and 4,051 controls of Chinese ancestry were used in the GWAS discovery analysis (Fig. 1 and Supplementary Table 1). The Manhattan plot of P-values using Cochran–Armitage trend test with adjustment for gender is shown in Supplementary Fig. 1. All cases and controls were assessed by principal components analysis for population stratification and were confirmed to be of Chinese ancestry (Supplementary Fig. 2). Quantile–quantile plots were constructed and genomic control values were calculated ($\lambda_{GC} = 1.04$) (Supplementary Fig. 3). Both of these results indicate negligible inflation of the genome-wide association signals caused by population stratification, further suggesting that the deviated tail of the P-values' distribution reflects some true genetic associations with NSCLP. We then conducted logistic regression analysis to assess the genotype–phenotype association.

To perform a fast-track replication study, we selected and genotyped 152 SNPs ($P < 1 \times 10^{-4}$) within 79 loci for a follow-up analysis in an additional 1,346 NSCLP cases and 4,542 controls of Chinese ancestry (cohort 2). Of the 146 successfully genotyped SNPs, 64 showed nominal association ($P < 0.05$ using logistic regression) in the validation stage and 61 of them showed a consistent direction in their estimated effects on risk between the discovery (cohort 1) and validation (cohort 2) stages (Supplementary Table 2). A fixed-effects meta-analysis of the combined cohorts 1 and 2, totalling 3,379 NSCLP cases and 8,593 controls, identified 14 new loci (20 SNPs) ($P < 5.00 \times 10^{-8}$ using Cochran–Mantel–Haenszel test), namely 2p25.1, 4p16.2, 4q28.1, 5p12, 6p24.3, 8p11.23, 8q22.1, 9q22.32, 12q13.13, 12q13.2, 12q21.1, 14q22.1, 14q32.13 and 17q21.32, and three suggestive loci 2q35, 8q22.2 and 20q13.2 (Table 1, Fig. 2 and Supplementary Table 3). We also confirmed 12 reported loci (21 SNPs): 1p22.1, 1q32.2, 2p24.2, 8q21.3, 8q24.21, 9q22.2, 10q25.3, 13q31.1, 16p13.3, 17p13.1, 17q22 and 20q12 ($P < 5.00 \times 10^{-8}$) (Fig. 2 and Supplementary Table 4). All these 26 susceptibility loci collectively account for 10.94% of the NSCLP heritability. In addition, conditional analyses were performed for all 26 loci and we identified a secondary signal in one previously reported locus at 1q32.2 (Supplementary Table 5). After reviewing the published GWASs of NSCL/P and the present study, we summarize the susceptibility loci identified in different populations in Supplementary Fig. 4.

**Replications of the 26 loci in sub-phenotype groups of NSOFC.** We successfully genotyped 40 of the 41 SNPs (1 SNP, rs481931 at 1p22.1, was unsuccessfully genotyped) from the 26 loci in cohort 3 (NSCLO) and cohort 4 (NSCPO). Two novel (14q32.13 and 17q21.32) and eight reported loci (1p22.1, 1q32.2, 2p24.2, 8q21.3, 9q22.2, 10q25.3, 17p13.1 and 20q12) showed significant associations ($P_{\text{Bonferroni}} < 1.25 \times 10^{-3}$ using logistic regression test and Bonferroni correction; 0.05 out of 40) with NSCLO (Supplementary Table 6). All the associated SNPs from the above ten loci have concordant associations in the effect sizes and direction in both NSCLP and NSCLO (Supplementary Tables 3, 4 and 6). Two loci (13q31.1 and 15q13.3) were reported to be more strongly associated with NSCLP than NSCLO[7–9]. We also found rs9545308 at 13q31.1 to be significantly associated with NSCLP ($P_{\text{NSCLP meta}} = 2.00 \times 10^{-9}$, odds ratio (OR) = 1.29) but not with NSCLO ($P_{\text{NSCLO}} = 4.95 \times 10^{-3}$, OR = 1.23) in our Chinese samples. The marker at 15q13.3 was not successfully replicated in NSCLP and thus was not chosen to be replicated in NSCLO in our study.

One novel (9q22.32) and two reported loci (1q32.2 and 8q24.21) showed significant associations with NSCPO (Supplementary Table 7). The marker in 1q32.2 showed opposite directions of association between the NSCLP and NSCPO groups (rs9430019; $\text{OR}_{\text{NSCLP}} = 1.25$ and $\text{OR}_{\text{NSCLO}} = 0.66$), whereas the markers in the 8q24.21 and 9q22.32 loci were concordant in the estimated direction of association with NSCLP (Supplementary Tables 3, 4 and 7). It is worth mentioning that the recent GWAS[17] and sequencing study[18] revealed an aetiological missense variant in GRHL3 (1p36.11) for NSCPO. The additional locus 9q22.33 (FOXE1) was identified potentially accounting for linkage to both NSCL/P and NSCPO[19]. The markers at 1p36.11 and 9q22.33 were not significant at the GWAS stage in our study and thus were not replicated in NSCLP and NSCPO.

Tests for heterogeneity showed that the SNPs at 1, 8 and 5 loci yielded significant evidence of heterogeneity ($P < 1.25 \times 10^{-3}$ using logistic regression test and Bonferroni correction; 0.05 out of 40) between NSCLO and NSCLP, NSCPO and NSCLP, and NSCPO and NSCLO, respectively (Table 2 and Table 3).

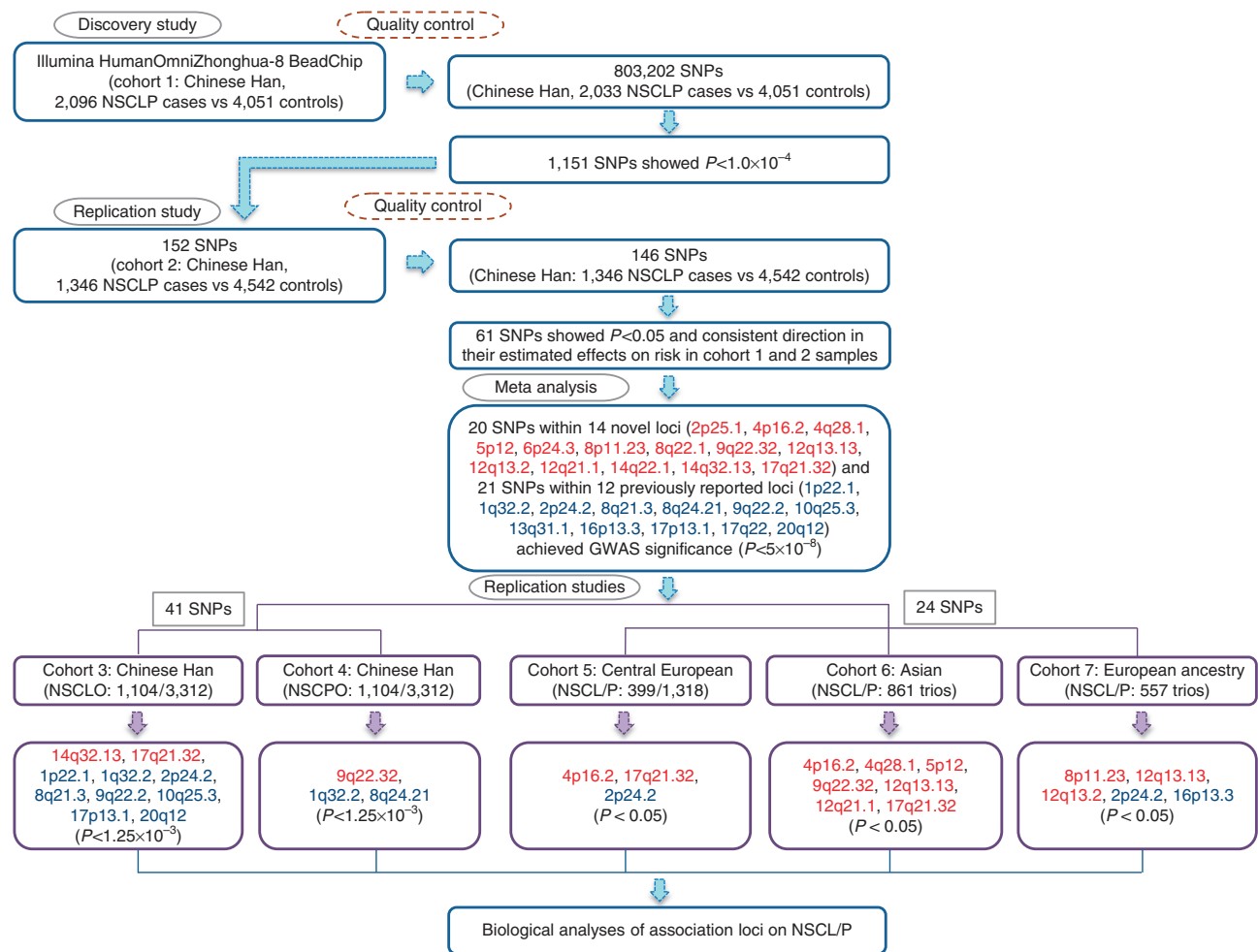

**Figure 1 | Study design.** We first conducted a GWAS study in 2,096 Chinese NSCLP cases and matched 4,051 controls using Illumina HumanOmniZhonghua-8 BeadChip. After quality control, 803,202 SNPs were remained and analysed in 2,033 NSCLP cases and 4,051 controls, and 1,151 SNPs showed $P<1.0\times10^{-4}$ using logistic regression in the discovery stage. One hundred and fifty-two SNPs with $P<1.0\times10^{-4}$ were selected for replication in an independent Chinese cohort including 1,346 NSCLP cases and 4,542 controls. After quality control, 146 SNPs remained, of which 61 SNPs showed $P<0.05$ using logistic regression and consistent direction in their estimated effects on risk in the discovery and validation samples. Then, a fixed-effects meta-analysis of the combined cohorts 1 and 2 samples identified 14 novel loci (20 SNPs) and confirmed 12 previously reported ones (21 SNPs) associated at genome-wide significance ($P_{meta}<5\times10^{-8}$ using Cochran–Mantel–Haenszel test). We genotyped 41 top SNPs in further 1,104 NSCLO, 1,104 NSCPO patients and 3,312 shared controls in Chinese Han population, respectively. As a result, ten and three loci showed significant associations in cohort 3 and 4 samples ($P_{Bonferroni}<1.25\times10^{-3}$ using logistic regression and Bonferroni correction). The 24 SNPs (20 from the 14 novel loci and 4 from two newly reported NSCL/P loci) out of the 41 SNPs were also evaluated in Central European, Asian and European ancestry populations, and 3, 7 and 5 loci showed evidence of association in the different cohorts, respectively ($P<0.05$ using logistic regression). We additionally explored the molecular functionalities of risk variants and their related genes using several complementary methods.

Interestingly, gender stratification analysis revealed that one previously identified locus (1q32.2) showed strong evidence of heterogeneity ($P=1.38\times10^{-4}$) in the evidence of association with NSCLP from male and female cases. The marker on 8q21.3 was observed to exhibit significant evidence of heterogeneity in its estimated effect between older mothers ($>35$ years) and the reference age of mothers (25–35 years) (Supplementary Table 8).

**Replications of 16 NSCLP loci in multi-ethnic groups.** We further checked for associations of the 14 novel loci and two recently reported NSCL/P loci (16p13.3 in China[15] and 2p24.2 in a multi-ethnic study[16]) using cohorts 5–7. Different loci showed evidence of association in different cohorts ($P<0.05$ using logistic regression test), specifically three loci in Central Europeans, seven loci in Asians and five loci in European ancestry (Table 1 and Supplementary Table 9). For the majority of the 16 loci mentioned, the direction and magnitude of the effect of ORs were

consistent across Chinese and non-Chinese samples. However, we observed an apparent difference in risk allele frequencies (AFs) for most of these 16 risk loci. For example, AFs in the cases of the markers at 4p16.2 (rs1907989, $AF_{Chinese}=0.46$, $AF_{European}=0.57$) and 17q21.32 (rs1838105, $AF_{Chinese}=0.45$, $AF_{European}=0.39$) showed a certain degree of difference between the Chinese and Central European populations (Table 1), whereas the AFs of the markers at 8q24.21 (rs987525, $AF_{Chinese}=0.07$, $AF_{European}=0.38$ and rs7017252, $AF_{Chinese}=0.08$, $AF_{European}=0.55$) were highly different between the Chinese and European populations (Supplementary Table 10).

**Biological implications analyses for the 26 NSCLP loci.** Of the 135 SNPs associated with the risk of NSCLP at these 26 loci ($r^2\geq0.7$ with the index SNPs), 33, 99 and 113, respectively, were found in known or predicted regulatory elements such as promoters, enhancers or motifs biochemically characterized to

**Table 1 | Association results for markers from the 14 novel NSCLP risk loci at each stage.**

| loci | SNP | BP (hg19) | Alleles* | NSCLP GWAS (above) and NSCLP replication (below) | | | | NSCLP Meta | | NSCLO replication (above) and NSCPO replication (below) | | | | NSCL/P replication 1 | | | | NSCL/P replication 2 (above) and NSCL/P replication 3 (below) | | | 1000 Genomes | |
|---|---|---|---|---|---|---|---|---|---|---|---|---|---|---|---|---|---|---|---|---|---|---|
| | | | | F_A | F_U | P | OR | P(F) | OR(F) | F_A | F_U | P | OR | F_A | F_U | P | OR | MAF | P | RR | MAF_CHB | MAF_EUR |
| 2p25.1 | rs287980 | 9971366 | G/A | 0.23 | 0.27 | 1.99E−05 | 0.82 | **1.94E−08** | 0.83 | 0.25 | 0.26 | 2.34E−01 | 0.93 | 0.20 | 0.21 | 5.34E−01 | 0.94 | 0.26 | 4.63E−01 | 0.94 | 0.28 | 0.22 |
| | | | | 0.23 | 0.27 | 5.54E−04 | 0.84 | | | 0.27 | 0.26 | 7.87E−01 | 1.02 | | | | | 0.21 | 7.62E−01 | 0.97 | | |
| 2p25.1 | rs287982 | 9972442 | G/A | 0.23 | 0.27 | 1.58E−05 | 0.82 | **6.15E−09** | 0.82 | 0.25 | 0.26 | 1.65E−01 | 0.92 | 0.20 | 0.21 | 5.21E−01 | 0.94 | 0.26 | 4.63E−01 | 0.94 | 0.28 | 0.22 |
| | | | | 0.23 | 0.27 | 2.28E−04 | 0.83 | | | 0.27 | 0.26 | 7.63E−01 | 1.02 | | | | | 0.21 | 6.85E−01 | 0.96 | | |
| 4p16.2 | rs34246903 | 4794195 | C/A | 0.38 | 0.43 | 2.57E−05 | 0.82 | **4.45E−08** | 0.85 | 0.40 | 0.42 | 2.91E−01 | 0.95 | 0.30 | 0.34 | **2.39E−02** | 0.82 | 0.42 | **2.54E−02** | 0.85 | 0.41 | 0.33 |
| | | | | 0.40 | 0.43 | 1.09E−02 | 0.89 | | | 0.41 | 0.42 | 7.77E−01 | 0.99 | | | | | 0.33 | 1.65E−01 | 1.13 | | |
| 4p16.2 | rs1907989 | 4818925 | A/G | 0.46 | 0.51 | 2.29E−06 | 0.81 | **1.58E−08** | 0.85 | 0.48 | 0.50 | 6.42E−02 | 0.91 | 0.57 | 0.61 | **1.92E−02** | 0.83 | 0.49 | **3.91E−03** | 1.22 | 0.52 | 0.64 |
| | | | | 0.48 | 0.50 | 1.94E−02 | 0.90 | | | 0.50 | 0.50 | 5.78E−01 | 1.03 | | | | | 0.36 | 5.14E−01 | 1.06 | | |
| 4q28.1 | rs908822 | 124906257 | A/G | 0.10 | 0.07 | 6.13E−07 | 1.46 | **4.33E−08** | 1.31 | 0.09 | 0.08 | 7.64E−02 | 1.17 | 0.08 | 0.07 | 2.44E−01 | 1.20 | 0.08 | **7.14E−03** | 1.41 | 0.07 | 0.06 |
| | | | | 0.10 | 0.09 | 4.46E−02 | 1.16 | | | 0.09 | 0.08 | 5.21E−01 | 1.06 | | | | | 0.07 | 3.80E−01 | 1.15 | | |
| 5p12 | rs10462065 | 44068846 | A/C | 0.24 | 0.20 | 3.51E−07 | 1.26 | **1.12E−08** | 1.22 | 0.24 | 0.22 | 7.92E−02 | 1.11 | 0.11 | 0.11 | 7.28E−01 | 0.96 | 0.24 | **3.01E−02** | 1.19 | 0.19 | 0.10 |
| | | | | 0.23 | 0.20 | 2.04E−03 | 1.18 | | | 0.22 | 0.22 | 7.17E−01 | 0.98 | | | | | 0.12 | 6.66E−01 | 1.05 | | |
| 6p24.3 | rs9381107 | 9469238 | G/A | 0.31 | 0.37 | 1.97E−09 | 0.79 | **2.72E−09** | 0.83 | 0.32 | 0.35 | 4.79E−02 | 0.90 | 0.16 | 0.18 | 1.43E−01 | 0.85 | 0.32 | 8.43E−01 | 1.02 | 0.33 | 0.16 |
| | | | | 0.33 | 0.36 | 1.40E−02 | 0.89 | | | 0.34 | 0.35 | 2.81E−01 | 0.95 | | | | | 0.15 | 7.75E−01 | 0.97 | | |
| 8p11.23 | rs13317 | 38269514 | G/A | 0.30 | 0.34 | 4.10E−06 | 0.83 | **3.96E−08** | 0.85 | 0.31 | 0.33 | 7.59E−02 | 0.91 | 0.21 | 0.22 | 4.46E−01 | 0.93 | 0.31 | 2.65E−01 | 0.92 | 0.39 | 0.25 |
| | | | | 0.32 | 0.35 | 7.43E−03 | 0.88 | | | 0.32 | 0.33 | 3.83E−01 | 0.95 | | | | | 0.23 | **3.17E−03** | 0.76 | | |
| 8q22.1 | rs12681366 | 95401265 | G/A | 0.44 | 0.49 | 2.95E−06 | 0.82 | **2.35E−10** | 0.83 | 0.46 | 0.48 | 6.16E−02 | 0.91 | 0.32 | 0.34 | 3.01E−01 | 0.92 | 0.45 | 5.62E−01 | 0.96 | 0.48 | 0.35 |
| | | | | 0.44 | 0.48 | 2.48E−04 | 0.85 | | | 0.47 | 0.48 | 4.01E−01 | 0.96 | | | | | 0.31 | 2.28E−01 | 0.90 | | |
| 8q22.1 | rs957448 | 95541302 | G/A | 0.44 | 0.50 | 3.78E−10 | 0.78 | **9.60E−13** | 0.81 | 0.47 | 0.49 | 5.72E−02 | 0.91 | 0.23 | 0.26 | 1.49E−01 | 0.87 | 0.47 | 1.18E−01 | 0.90 | 0.51 | 0.31 |
| | | | | 0.46 | 0.49 | 7.59E−04 | 0.86 | | | 0.48 | 0.49 | 1.33E−01 | 0.93 | | | | | 0.23 | 4.07E−01 | 0.92 | | |
| 9q22.32 | rs10512248 | 98259703 | C/A | 0.29 | 0.34 | 3.78E−07 | 0.79 | **5.10E−10** | 0.82 | 0.33 | 0.35 | 4.47E−02 | 0.90 | 0.34 | 0.33 | 4.34E−01 | 1.07 | 0.28 | **1.41E−02** | 0.81 | 0.31 | 0.32 |
| | | | | 0.30 | 0.33 | 3.19E−03 | 0.87 | | | 0.31 | 0.35 | **1.23E−04** | 0.82 | | | | | 0.32 | 1.00E−01 | 0.86 | | |
| 12q13.13 | rs3741442 | 53346750 | G/A | 0.46 | 0.41 | 1.17E−06 | 1.22 | **3.72E−12** | 1.22 | 0.44 | 0.42 | 6.42E−02 | 1.10 | 0.99 | 1.00 | 5.42E−01 | 0.72 | 0.54 | **1.30E−04** | 0.77 | 0.41 | 0.98 |
| | | | | 0.46 | 0.41 | 3.36E−06 | 1.23 | | | 0.44 | 0.42 | 9.72E−02 | 1.09 | | | | | 0.02 | **2.52E−02** | 0.48 | | |
| 12q13.2 | rs705704 | 56435412 | A/G | 0.27 | 0.23 | 4.39E−05 | 1.22 | **1.29E−09** | 1.22 | 0.24 | 0.25 | 2.84E−01 | 0.89 | 0.34 | 0.31 | 7.44E−01 | 1.16 | 0.23 | 6.61E−01 | 1.04 | 0.24 | 0.33 |
| | | | | 0.28 | 0.24 | 5.97E−06 | 1.22 | | | 0.25 | 0.25 | 1.00E+00 | 1.00 | | | | | 0.35 | **1.15E−03** | 1.32 | | |
| 12q21.1 | rs2304269 | 72080272 | G/A | 0.38 | 0.44 | 1.07E−07 | 0.79 | **1.32E−12** | 0.81 | 0.41 | 0.44 | 2.04E−02 | 0.89 | 0.05 | 0.05 | 7.81E−01 | 0.95 | 0.43 | **2.66E−02** | 0.85 | 0.55 | 0.06 |
| | | | | 0.41 | 0.45 | 1.12E−04 | 0.84 | | | 0.45 | 0.44 | 3.03E−01 | 1.05 | | | | | 0.07 | 6.17E−01 | 0.92 | | |
| 12q21.1 | rs7967428 | 72089040 | G/A | 0.38 | 0.44 | 1.63E−07 | 0.79 | **3.08E−12** | 0.81 | 0.41 | 0.44 | 6.56E−02 | 0.91 | 0.05 | 0.05 | 9.91E−01 | 1.00 | 0.43 | **2.32E−02** | 0.85 | 0.55 | 0.06 |
| | | | | 0.41 | 0.45 | 1.57E−04 | 0.84 | | | 0.45 | 0.44 | 1.72E−01 | 1.07 | | | | | 0.07 | 6.78E−01 | 0.93 | | |
| 14q22.1 | rs7148069 | 51839645 | A/G | 0.23 | 0.19 | 1.94E−05 | 1.27 | **1.69E−08** | 1.22 | 0.21 | 0.20 | 6.21E−01 | 1.03 | 0.32 | 0.33 | 7.92E−01 | 0.98 | 0.17 | 8.75E−01 | 1.02 | 0.19 | 0.33 |
| | | | | 0.22 | 0.20 | 7.21E−03 | 1.16 | | | 0.19 | 0.20 | 4.90E−01 | 0.96 | | | | | 0.31 | 3.06E−01 | 1.11 | | |
| 14q32.13 | rs1243572 | 95379499 | G/A | 0.48 | 0.42 | 1.37E−07 | 1.26 | **3.52E−10** | 1.20 | 0.47 | 0.43 | **7.75E−04** | 1.18 | 0.79 | 0.78 | 3.65E−01 | 1.09 | 0.57 | 7.84E−01 | 0.98 | 0.40 | 0.79 |
| | | | | 0.45 | 0.42 | 6.09E−03 | 1.13 | | | 0.44 | 0.43 | 2.14E−01 | 1.06 | | | | | 0.23 | 4.99E−01 | 0.94 | | |
| 14q32.13 | rs1243573 | 95379583 | C/A | 0.48 | 0.42 | 2.83E−07 | 1.25 | **8.61E−10** | 1.20 | 0.46 | 0.43 | 8.19E−03 | 1.14 | 0.80 | 0.78 | 4.10E−01 | 1.09 | 0.57 | 7.84E−01 | 0.98 | 0.40 | 0.79 |
| | | | | 0.45 | 0.42 | 7.30E−03 | 1.13 | | | 0.45 | 0.43 | 8.60E−02 | 1.09 | | | | | 0.23 | 4.99E−01 | 0.94 | | |
| 17q21.32 | rs4968247 | 44988703 | A/G | 0.37 | 0.41 | 1.94E−06 | 0.84 | **8.70E−10** | 0.83 | 0.37 | 0.40 | 1.44E−01 | 0.85 | 0.66 | 0.67 | 4.81E−01 | 0.94 | 0.61 | **2.52E−03** | 1.26 | 0.43 | 0.65 |
| | | | | 0.37 | 0.41 | 2.43E−06 | 0.83 | | | 0.40 | 0.40 | 4.34E−01 | 0.96 | | | | | 0.33 | 1.21E−01 | 1.15 | | |
| 17q21.32 | rs1838105 | 45008935 | A/G | 0.45 | 0.39 | 1.84E−07 | 1.26 | **1.31E−11** | 1.22 | 0.43 | 0.38 | **5.98E−04** | 1.19 | 0.39 | 0.33 | **5.83E−03** | 1.27 | 0.40 | **1.43E−02** | 1.19 | 0.50 | 0.37 |
| | | | | 0.42 | 0.38 | 3.48E−04 | 1.17 | | | 0.38 | 0.38 | 9.30E−01 | 1.00 | | | | | 0.38 | 1.37E−01 | 1.13 | | |

F_A, allele frequency in cases; F_U, allele frequency in controls; GWAS. Genome-wide association study; NSCLO, non-syndromic cleft lip only; NSCLP, non-syndromic cleft lip with palate; NSCPO, non-syndromic cleft palate only; SNP, single-nucleotide polymorphism.
NSCLP GWAS: study in 2,033 NSCLP cases and 4,051 controls of Chinese Han (the $P$-value using logistic regression test); NSCLP replication: replication study in 1,346 NSCLP cases and 4,542 controls of Chinese Han (the $P$-value using logistic regression test); NSCLP Meta: Meta analysis of NSCLP GWAS and NSCLP replication using fixed model (the $P$-value using Cochran–Mantel–Haenszel test); NSCLO replication: replication study in 1,104 NSCLO cases and 3,312 controls of Chinese Han (the $P$-value using logistic regression test); NSCPO replication: replication study in 1,104 NSCPO cases and 3,312 controls shared with NSCLO replication of Chinese Han (the $P$-value using logistic regression test); NSCL/P replication 1: replication study in 399 NSCL/P cases and 1,318 controls of Central Europeans (the $P$-value using logistic regression test); NSCL/P replication 2: replication study in 861 NSCL/P case-parent trios of Asian ancestry (the $P$-value using logistic regression test); NSCL/P replication 3: replication study in 557 NSCL/P case–parent trios of European American ancestry (the $P$-value using logistic regression test).
*Alleles: shown as minor allele/major allele in Chinese Han samples. Bold denotes entries that reached the thresholds of significance.

regulate transcription for 50 reference genes (Supplementary Data 1). Expression profiles of the 49 genes within 500 kb and showing strong linkage disequilibrium (LD) with index SNPs ($r^2 \geq 0.7$) in the 26 loci based on craniofacial gene expression patterns in mouse (EMAGE database) showed that a total of these 20 genes were related to embryonic development, among which 12 were expressed in the related tissue for OFC (Supplementary Table 11). Twelve genes were found to produce CL/CP malformations in mutant mouse models (Supplementary Table 12). Seven genes were reported to be associated with nine recognized malformation syndromes including OFC as a clinical phenotype (Supplementary Table 13). By manually reviewing the literature related to these 49 genes and their functions, annotated systematically using several databases, 28 of these genes were classified into different biological categories according to morphogenesis, development, molecular and cellular function, and 17 of these categories contained 3 or more genes. The most relevant catalogue was the morphogenesis- and development-related traits, which included the bone, limb, brain, ear and other organs. Several important signalling pathways were also reviewed, among which epidermis/epithelium development and morphogenesis, and the fibroblast growth factor receptor (FGFR) signalling pathway involved the most genes, whereas the bone morphogenic proteins (BMP), WNT and Notch signalling pathways have also been reported to be associated with lip and palate development (Supplementary Table 14).

## Discussion

In this study, we identified 14 novel risk loci associated with NSCLP and confirmed 12 previously reported loci in the Chinese population. Several lines of evidence support the hypothesis that these newly identified loci contribute to orofacial clefting. At 4p16.2, two markers in moderate LD with one another ($r^2 = 0.65$) are located 50 kb upstream of *MSX1* (Supplementary Fig. 5b). Mutations in *MSX1* contribute to CL/P and tooth agenesis in humans[20]. *Msx1*$^-$ homozygotic mice exhibit CP, deficiency of the alveolar mandible and maxilla, and failure of tooth development[21]. The orthologous *Spry1* of the optimized gene *SPRY1* at 4q28.1 has been shown to exhibit facial clefting and CP in transgenic mice[22]. The marker at 5p12 is adjacent to *FGF10*, whose encoded protein has been reported to play roles in the epithelial–mesenchymal transition, and *Fgf10* knockout mice show the symptoms of CP[23]. The 6p24.3 region harbours two candidate genes for the OFC: *OFCC1* was reported as a potential OFC susceptibility gene in humans, based on the observation of three unrelated OFC patients all harbouring a chromosomal break within or close to the gene[24]. *TFAP2A* mutations have been found in patients diagnosed with branchio-oculo-facial syndrome, which is characterized by branchial defects, ocular anomalies and facial defects, including CL/P[25].

The SNP at 8p11.23 is located in an enhancer within *FGFR1*. In humans, mutations of *FGFR1* have been found in syndromic CL/P[26], whereas in mice a homozygous hypomorphic allele at *Fgfr1* caused CP[27]. A marker at 9q22.32 is located at the intronic region of *PTCH1*, mutations of which have been identified in patients of Gorlin syndrome with CL/P as one of its clinical manifestations[28]. *Ptch1*$^{DL}$ mouse models showed craniofacial defects, including underdeveloped palatal shelves and clefting of the secondary palate[29]. The marker rs705704 is located at the locus 12q13.2, which includes the gene *RPS26*, which may be relevant to orofacial clefting, as mutations of *RPS26* have been reported in Diamond–Blackfan anaemia (DBA) with CPO as one

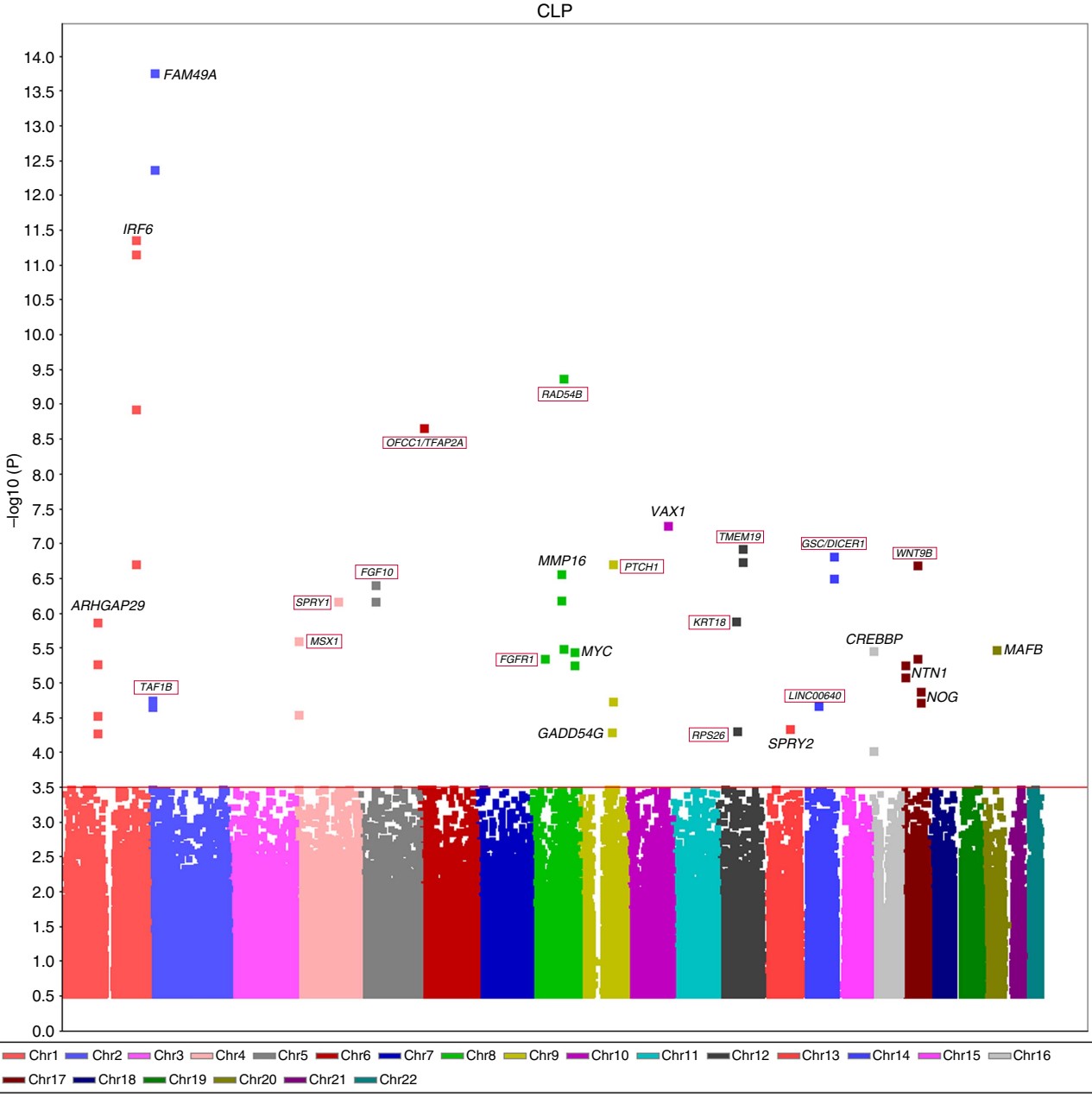

**Figure 2 | Manhattan plot of the association evidence of the 26 NSCLP risk loci in the discovery stage.** Prioritized genes from the 14 novel loci are encircled with red box, the remaining candidate genes are from the 12 previously reported loci.

of its clinical phenotypes[30]. Two highly correlated markers ($r^2 = 0.99$) at 14q32.13 are located between *GSC* and *DICER1*. *GSC* modulates the epithelial–mesenchymal transition and mutations in *GSC* lead to a syndrome defined by short stature, auditory canal atresia, mandibular hypoplasia and skeletal abnormalities[31], whereas *DICER1* mutations have been reported to cause pleuropulmonary blastoma and multinodular goiter-1, with or without Sertoli–Leydig cell tumours (MNG1)[32], and *Dicer1* conditional knockout mice exhibit secondary palate clefting and other severe craniofacial dysmorphisms[33]. The excellent candidate gene *WNT9B* at 17q21.32 has already been functionally implicated in craniofacial development, as mice with *Wnt9b* targeted mutation were described as presenting CL/P phenotypes[34].

Among the other new signals, two markers at 2p25.1 are in perfect LD ($r^2 = 1$) with one another and are located 12 kb

upstream of *TAF1B*, encoded protein of which is important for polymerase (Pol) I transcription[35]. At 8q22.1, a synonymous codon SNP rs957448 (*KIAA1429*) is correlated ($r^2 = 0.65$) with rs12681366 (an intronic SNP of *RAD54B*). Human *RAD54B* was first identified as a homologue of *RAD54*, which plays an important role in DNA repair[36]. The strongest associated marker at 12q13.13 is located 500 bp downstream of *KRT18*, which encodes a protein in the large family of cytoskeletal proteins with specific expression in epithelial cells[37]. At 12q21.1, the signals are near the *TMEM19* gene, involving the SNPs rs2304269 and rs7967428, which are in strong LD with each other ($r^2 = 0.98$). Rs2304269 and rs7967428 are respectively located at one active promoter and five strong enhancers in epidermal keratinocytes according to ENCODE data. The 14q22.1 signal is close to *LINC00640*, a gene of unknown function. In addition, to gain further insight into the possible involvement of genes at some

**Table 2 | SNPs showing significance in stratified analysis among the three anatomical types of orofacial clefts: NSCLP, NSCLO and NSCPO in Chinese population.**

| Phenotype | Loci | SNP | BP (hg19) | Allele | F_A | F_U | P* | OR |
|---|---|---|---|---|---|---|---|---|
| NSCLP versus NSCLO | 1q32.2 | rs861020 | 209977111 | A/G | 0.23 | 0.30 | 2.05E − 09 | 0.72 (0.65-0.80) |
| | 1q32.2 | rs642961 | 209989270 | A/G | 0.23 | 0.30 | 1.16E − 09 | 0.72 (0.64-0.80) |
| NSCLP versus NSCPO | 1q32.2 | rs861020 | 209977111 | A/G | 0.23 | 0.17 | 8.69E − 11 | 1.51 (1.34-1.72) |
| | 1q32.2 | rs642961 | 209989270 | A/G | 0.23 | 0.17 | 9.08E − 11 | 1.51 (1.33-1.72) |
| | 1q32.2 | rs2064163 | 210048819 | A/C | 0.38 | 0.44 | 6.41E − 08 | 0.76 (0.69-0.84) |
| | 1q32.2 | rs9430019 | 210050794 | A/G | 0.31 | 0.19 | 1.29E − 25 | 1.87 (1.66-2.11) |
| | 2p25.1 | rs287980 | 9971366 | G/A | 0.23 | 0.27 | 7.67E − 04 | 0.83 (0.74-0.92) |
| | 2p25.1 | rs287982 | 9972442 | G/A | 0.23 | 0.27 | 8.29E − 04 | 0.83 (0.74-0.93) |
| | 2p24.2 | rs10172734 | 16733054 | G/A | 0.26 | 0.33 | 6.31E − 09 | 0.73 (0.66-0.81) |
| | 2p24.2 | rs7552 | 16733928 | A/G | 0.24 | 0.28 | 6.33E − 04 | 0.83 (0.74-0.92) |
| | 8q21.3 | rs1034832 | 88918331 | C/A | 0.31 | 0.35 | 4.11E − 04 | 0.83 (0.75-0.92) |
| | 12q21.1 | rs2304269 | 72080272 | G/A | 0.39 | 0.45 | 8.58E − 07 | 0.78 (0.71-0.86) |
| | 12q21.1 | rs7967428 | 72089040 | G/A | 0.40 | 0.45 | 1.98E − 06 | 0.79 (0.72-0.87) |
| | 16p13.3 | rs2283487 | 3969886 | G/A | 0.42 | 0.47 | 3.21E − 04 | 0.84 (0.76-0.92) |
| | 16p13.3 | rs17136624 | 3996282 | A/G | 0.26 | 0.22 | 8.59E − 04 | 1.22 (1.08-1.36) |
| | 17p13.1 | rs2872615 | 8914693 | G/A | 0.43 | 0.49 | 2.59E − 07 | 0.78 (0.70-0.85) |
| | 17p13.1 | rs1880646 | 8929845 | G/A | 0.48 | 0.53 | 2.45E − 04 | 0.83 (0.76-0.92) |
| | 17q21.32 | rs1838105 | 45008935 | A/G | 0.44 | 0.38 | 2.28E − 05 | 1.24 (1.12-1.37) |
| NSCLO versus NSCPO | 1q32.2 | rs861020 | 209977111 | A/G | 0.30 | 0.17 | 3.98E − 24 | 2.10 (1.81-2.42) |
| | 1q32.2 | rs642961 | 209989270 | A/G | 0.30 | 0.17 | 2.09E − 24 | 2.11 (1.82-2.43) |
| | 1q32.2 | rs2064163 | 210048819 | A/C | 0.39 | 0.44 | 1.21E − 04 | 0.79 (0.70-0.89) |
| | 1q32.2 | rs9430019 | 210050794 | A/G | 0.30 | 0.19 | 8.70E − 18 | 1.84 (1.60-2.12) |
| | 2p24.2 | rs10172734 | 16733054 | G/A | 0.25 | 0.33 | 6.20E − 09 | 0.68 (0.59-0.77) |
| | 8q21.3 | rs1034832 | 88918331 | C/A | 0.28 | 0.35 | 1.13E − 06 | 0.73 (0.64-0.83) |
| | 10q25.3 | rs6585429 | 118893231 | G/A | 0.38 | 0.43 | 1.11E − 03 | 0.82 (0.72-0.92) |
| | 17p13.1 | rs2872615 | 8914693 | G/A | 0.42 | 0.49 | 1.34E − 05 | 0.77 (0.68-0.86) |

F_A, minor allele frequency in cases; F_U, minor allele frequency in controls; NSCLO, non-syndromic cleft lip only; NSCLP, non-syndromic cleft lip with palate; NSCPO, non-syndromic cleft palate only; OR, odds ratio; SNP, single-nucleotide polymorphism.
OR is calculated based on minor allele; alleles are shown as minor allele/major allele.
*P-value below $1.25 \times 10^{-3}$ (0.05out of 40, the P-value using logistic regression test and Bonferroni correction) was considered to be statistically significant.

identified loci in the development of NSCLP, immuno-histochemistry (IHC) analysis performed in mice at different embryonic stages found positive IHC staining of three genes of interest (*Rad54b*, *Rps26* and *Fam49a*) in the palatal mesenchymal cells and epithelium cells (Supplementary Fig. 6).

Notably, in our study, two members of the FGF signalling pathway, including *FGF10* at 5p12 and *FGFR1* at 8p11.23, as well as three FGF signalling regulatory genes (*SPRY1* (ref. 38) at 4q28.1, *PTCH1* (ref. 39) at 9q22.32 and *WNT9B* (ref. 40) at 17q21.32) were found to be associated with the risk of NSCLP. We performed a network analysis of notable genes in the 26 NSCLP associated loci, which showed that several FGF signalling related genes such as *FGFR1* and *FGF10* are connected (Fig. 3). The FGF signalling pathway was proposed to contribute to NSCL/P[41] and previous candidate gene studies have provided evidence in humans and animal models[41,42]. The findings of our association study strengthen the hypothesis that the FGF signalling pathway might play important roles in craniofacial development. Intriguingly, we also found a potential link between ribosomopathies and the genes in our NSCLP-associated loci, including *RPS26*, *RAD54B* and *TAF1B*. Mutations in *RPS26* were reported to affect the functions of the proteins in ribosomal RNA processing in DBA patients and DBAs belong to a class of diseases called ribosomopathies[30,43]. Moreover, *RPS26* and *RAD54B* were reported to regulate *p53* (refs 44,45) and the p53 pathway is importantly involved in ribosome biogenesis[43]. In addition, TAF1B was reported as a component of RNA Pol I basal transcription factor, which is essential for Pol I recruitment to the ribosomal RNA gene promoter[35].

For the 12 significant associated loci in the study that had been previously reported, the strongest signals occurred for 2 SNPs in near-perfect LD ($r^2 = 0.92$) in 2p24.2, located in the 3′-untranslated region of *FAM49A*. It is worth mentioning that both the LD block and ± 500 kb on either side of the index SNPs in this region only contain the single gene *FAM49A*, although a few non-coding RNA genes are located in this region. *FAM49A* is a protein-coding gene whose paralogue, *FAM49B*, is located in a previously reported susceptibility locus near the gene desert region of 8q24, which shows a strong association with the risk of NSCL/P in European populations[10,46,47]. ENCODE data indicate that SNP rs7552 alters the regulatory motifs of *TBX5* and *BRCA1*, and the highly correlated SNP rs4832651 ($r^2 = 0.98$) lies within a conserved enhancer for mammary epithelial cell activity. Although *Myc*-oncogene has been reported as the probable target effect gene in the 8q24 region for NSCL/P[47], the functions of both *FAM49A* and *FAM49B* remain poorly defined. These genes might play a role in the aetiology of NSCL/P and whether their functions vary across different populations is clearly worth further investigation. In addition, as expected, the second strongest signals were near *IRF6* at 1q32.2 and this association signal has been independently replicated in numerous GWAS studies and candidate gene studies[2,6,10,12,14–16]. Of the remaining ten loci, 1p36.13 and 3q12.1 approached genome-wide significance, 15q22.2 showed suggestive evidence of association and the additional seven loci were only analysed in the NSCLP GWAS stage (Supplementary Table 4).

Comparisons of NSCLP, NSCLO and NSCPO have yielded clear evidence of genetic heterogeneity among the three sub-groups of NSOFC. The two sub-groups (NSCLP and NSCLO) generally grouped together appeared to share more genetic risk factors, which is consistent with previous findings[4,6,48], and these results argue for distinct origins of development of the lip and primary palate versus the secondary palate[1,49]. In addition, although 1p36.11 and 9q22.33 were not confirmed in NSCPO in

**Table 3 | Markers achieving genome-wide significance in GWAS of 3,379 NSCLP cases and 8,593 controls of Chinese Han and prioritized genes in each significant SNP.**

| Loci | SNP | BP | Allele | $P_{Meta}$ | OR | $P_{het}$* | Notable gene(s) |
|---|---|---|---|---|---|---|---|
| *Novel loci:* | | | | | | | |
| 2p25.1 | rs287980 | 9971366 | G/A | 1.94E − 08 | 0.83 | 0.8120 | *TAF1B* |
| 2p25.1 | rs287982 | 9972442 | G/A | 6.15E − 09 | 0.82 | 0.8981 | *TAF1B* |
| 4p16.2 | rs34246903 | 4794195 | C/A | 4.45E − 08 | 0.85 | 0.2344 | *MSX1* |
| 4p16.2 | rs1907989 | 4818925 | A/G | 1.58E − 08 | 0.85 | 0.1130 | *MSX1* |
| 4q28.1 | rs908822 | 124906257 | A/G | 4.33E − 08 | 1.31 | 0.0545 | *SPRY1* |
| 5p12 | rs10462065 | 44068846 | A/C | 1.12E − 08 | 1.22 | 0.4835 | *FGF10* |
| 6p24.3 | rs9381107 | 9469238 | A/G | 2.72E − 09 | 0.83 | 0.0900 | *OFCC1/TFAP2A* |
| 8p11.23 | rs13317 | 38269514 | G/A | 3.96E − 08 | 0.85 | 0.4601 | *FGFR1* |
| 8q22.1 | rs12681366 | 95401265 | G/A | 2.35E − 10 | 0.83 | 0.5965 | *RAD54B* |
| 8q22.1 | rs957448 | 95541302 | G/A | 9.60E − 13 | 0.81 | 0.1260 | *RAD54B* |
| 9q22.32 | rs10512248 | 98259703 | C/A | 5.10E − 10 | 0.82 | 0.2026 | *PTCH1* |
| 12q13.13 | rs3741442 | 53346750 | G/A | 3.72E − 12 | 1.22 | 0.9598 | *KRT18* |
| 12q13.2 | rs705704 | 56435412 | A/G | 1.29E − 09 | 1.22 | 0.9839 | *RPS26* |
| 12q21.1 | rs2304269 | 72080272 | G/A | 1.32E − 12 | 0.81 | 0.3903 | *TMEM19* |
| 12q21.1 | rs7967428 | 72089040 | G/A | 3.08E − 12 | 0.81 | 0.3871 | *TMEM19* |
| 14q22.1 | rs7148069 | 51839645 | A/G | 1.69E − 08 | 1.22 | 0.2538 | *LINC00640* |
| 14q32.13 | rs1243572 | 95379499 | G/A | 3.52E − 10 | 1.20 | 0.1138 | *GSC/DICER1* |
| 14q32.13 | rs1243573 | 95379583 | C/A | 8.61E − 10 | 1.20 | 0.1178 | *GSC/DICER1* |
| 17q21.32 | rs4968247 | 44988703 | A/G | 8.70E − 10 | 0.83 | 0.8605 | *WNT9B* |
| 17q21.32 | rs1838105 | 45008935 | A/G | 1.31E − 11 | 1.22 | 0.3543 | *WNT9B* |
| | | | | | | | |
| *Reported loci:* | | | | | | | |
| 1p22.1 | rs481931 | 94570016 | A/C | 1.06E − 12 | 0.80 | 0.3687 | *ARHGAP29* |
| 1p22.1 | rs4147803 | 94582293 | G/C | 7.97E − 12 | 0.81 | 0.8369 | *ARHGAP29* |
| 1q32.2 | rs861020 | 209977111 | A/G | 1.30E − 14 | 1.31 | 0.5428 | *IRF6* |
| 1q32.2 | rs642961 | 209989270 | A/G | 2.76E − 15 | 1.32 | 0.6061 | *IRF6* |
| 1q32.2 | rs2064163 | 210048819 | A/C | 8.60E − 19 | 0.77 | 0.9625 | *IRF6* |
| 1q32.2 | rs9430019 | 210050794 | A/G | 1.68E − 12 | 1.25 | 0.6420 | *IRF6* |
| 2p24.2 | rs10172734 | 16733054 | G/A | 2.89E − 20 | 0.74 | 0.4992 | *FAM49A* |
| 2p24.2 | rs7552 | 16733928 | A/G | 5.83E − 22 | 0.73 | 0.5814 | *FAM49A* |
| 8q21.3 | rs12543318 | 88868340 | A/C | 8.80E − 12 | 0.81 | 0.2050 | *MMP16* |
| 8q21.3 | rs1034832 | 88918331 | C/A | 1.35E − 10 | 0.82 | 0.2243 | *MMP16* |
| 8q24.21 | rs7845615 | 129888794 | A/G | 1.03E − 10 | 1.27 | 0.8110 | *MYC* |
| 8q24.21 | rs7017252 | 129950844 | A/G | 8.47E − 16 | 1.60 | 0.8960 | *MYC* |
| 9q22.2 | rs7871395 | 92209587 | A/G | 6.06E − 09 | 1.21 | 0.5782 | *GADD45G* |
| 10q25.3 | rs6585429 | 118893231 | G/A | 7.14E − 13 | 0.81 | 0.9967 | *VAX1* |
| 13q31.1 | rs9545308 | 80639405 | A/C | 2.00E − 09 | 1.29 | 0.8103 | *SPRY2* |
| 16p13.3 | rs2283487 | 3969886 | G/A | 1.27E − 10 | 0.83 | 0.9121 | *CREBBP* |
| 16p13.3 | rs17136624 | 3996282 | A/G | 3.82E − 10 | 1.24 | 0.5269 | *CREBBP* |
| 17p13.1 | rs2872615 | 8914693 | G/A | 8.81E − 12 | 0.82 | 0.5224 | *NTN1* |
| 17p13.1 | rs1880646 | 8929845 | A/G | 1.69E − 11 | 1.22 | 0.4104 | *NTN1* |
| 17q22 | rs227731 | 54773238 | C/A | 8.83E − 09 | 1.19 | 0.5623 | *NOG* |
| 20q12 | rs6129653 | 39275603 | A/G | 8.57E − 12 | 1.23 | 0.5970 | *MAFB* |

GWAS, genome-wide association study; NSCLP, non-syndromic cleft lip with palate; OR, odds ratio; SNP, single-nucleotide polymorphism.
Genome-wide significance is defined as $P < 5 \times 10^{-8}$; SNP positions are reported according to Build 37 and their alleles are coded based on the positive strand; alleles (minor/major); meta-analysis is of NSCLP GWAS and NSCLP replication using fixed model; the $P$-value using Cochran–Mantel–Haenszel test; OR is calculated based on minor allele.
*$P_{het}$: $P$-value for heterozygosity test using logistic regression test and Bonferroni correction and $P_{het} > 0.05$ was considered to signify no heterogeneity.

our study, 1q32.2, 8q24.21 and 9q22.32 were first demonstrated to have an effect on NSCPO in the Chinese population. Importantly, our study provided evidence that 1q32.2 exhibits an overlapping effect on all three sub-phenotypes of NSCLP, NSCLO and NSCPO. The evidence of association at 1q32.2 was stronger among males than females, which may reflect the higher prevalence rate of NSCLP among males (male:female = 2.6:1 in our study). Stratification of the results by maternal gestational age revealed that older mothers may have a higher risk of having a child with NSCLP, as suggested by some previous studies of congenital disorders such as NSOFC[50,51].

Plausible reasons for the failure to replicate all of the associated loci in different ethnic groups could be due to the limited sample size, the combined sub-groups of NSCL/P used in previous analyses, the differential tagging of unobserved causal variants across ethnic groups or the existence of true genetic heterogeneity

across ethnic groups. Further studies using larger sample sizes or analytical approaches, such as a *trans*-ethnic genome-wide meta-analysis approach[52] with more detailed classification of sub-phenotypes, are warranted to further investigate this hypothesis.

Of the 26 genetic risk factors, 19 had reported associations with a total of 34 other diseases/traits. These associations could mainly be categorized into six different groups, including developmental, immune, metabolic, neoplastic, endocrine and degenerative categories (Supplementary Data 2). To further assess the possible independence among these various birth defects/diseases/traits within these particular SNPs, we examined LD patterns between these SNPs in Asian, African and European populations using data from the 1,000 Genomes Project. As a result, three susceptibility loci were identified to be shared by NSCLP and other diseases/traits, including schizophrenia at 8p11.23, asthma, polycystic ovary syndrome, rheumatoid arthritis, vitiligo, type 1

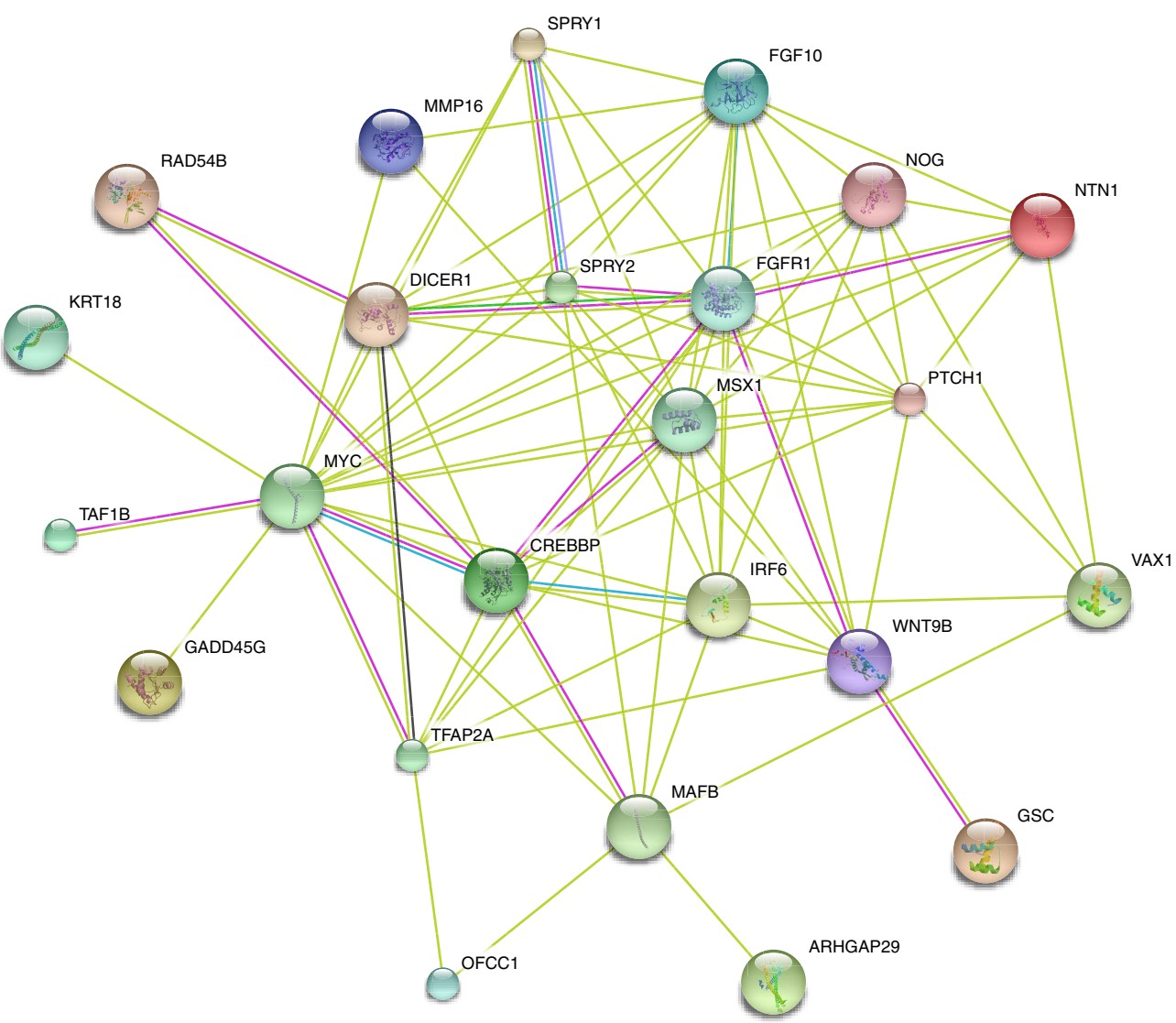

**Figure 3 | Functional similarity network of genes in the 26 NSCLP associated loci.** The network in this figure is constructed for the 28 notable genes from the present study. Genes and their nodes that are not connected to any other node in the network are omitted. Thus, 24 out of the 28 genes are left and highly involved in the pathway network. The network nodes are proteins. The edges represent the predicted functional associations. An edge may be drawn with up to seven differently coloured lines—these lines represent the existence of the seven types of evidence used in predicting the associations. A red line indicates the presence of fusion evidence; a green line—neighbourhood evidence; a blue line—coocurrence evidence; a purple line—experimental evidence; a yellow line—textmining evidence; a light blue line—database evidence; a black line—coexpression evidence.

diabetes autoantibodies and alopecia at 12q13.2, and height at 9q22.32. The SNPs reported to be associated with NSCLP and other diseases/traits were in strong LD ($r^2 \geq 0.7$), which showed significant and non-independent association of the risk of NSCLP and other diseases/traits. Interestingly, NSCLP and adult height shared the same index SNP (rs10512248 at *PTCH1* in 9q22.32), suggesting that some shared genetic factors might underlie these two very distinct phenotypes. Furthermore, some reported GWAS loci had susceptibility genes shared between NSCLP and other diseases/traits, such as MAFB at 20q12 for Dupuytren's disease, low-density lipoprotein cholesterol and total cholesterol, which showed a clearly independent association with NSCLP index SNPs and suggested pleiotropic effects of these genes on other diseases/traits.

Overall, our current study has advanced the understanding of the genetic architecture controlling the risk of NSOFC by substantially increasing the number of genetic risk factors and

has highlighted potential candidate genes through subsequent genetic and biological analyses. This study has also provided further insight into the possible pleiotropic effects of genetic risk factors on different sub-phenotypes, in different populations and among different diseases/traits. Through a comprehensive analysis of cases and controls from a Chinese population, we have identified 14 new genetic risk factors and validated associations in a large majority of previously reported loci. Further sequencing and functional investigations will probably identify causal mutational events and true susceptibility genes in or near these tagging SNPs and further elucidate the disease pathogenesis of these common congenital birth defects.

## Methods
**Samples.** In the current study, we carried out a two-stage GWAS and further replications of NSOFC. The discovery stage included 2,096 NSCLP cases and 4,051 controls (cohort 1). Replication studies were performed in an additional 1,346

unrelated NSCLP cases and 4,542 controls (cohort 2). Further replications consisted of cohort 3 (1,104 NSCLO cases versus 3,312 controls), cohort 4 (1,104 NSCPO cases versus 3,312 controls shared with cohort 3), cohort 5 (399 NSCL/P cases versus 1,318 controls), cohort 6 (861 NSCL/P case–parent trios) and cohort 7 (557 NSCL/P case–parent trios). Samples of cohorts 1–4 were recruited from the Chinese population through collaboration with multiple hospitals in Hubei, Henan and Anhui province. All cases were interviewed and clinically assessed by at least two experienced clinicians, and a detailed questionnaire was completed to identify any further anomalies, such as congenital heart disease, hypospadias, accessory auricle, lip pits and so on, which would suggest an underlying syndrome. We collected clinical information from the subjects through a full clinical checkup and additional demographic information from the cases was obtained through a structured questionnaire that mainly included four parts: basic information, clinical feature, maternal situation and life style during the first trimester of pregnancy, and genetic background of the patients. All controls were healthy individuals without OFC or family history of OFC (including first-, second- and third-degree relatives). Peripheral blood samples were collected after the written informed consents were obtained from all the participants or their guardians. The study was approved by the institutional ethics committee of each hospital (Hospital of Stomatology Wuhan University, The Second Charity Hospital of Henan Province, Stomatological Hospital of Nanyang, Stomatological Hospital of Xiangyang and The First Affiliated Hospital of Anhui Medical University) and was conducted according to the Declaration of Helsinki principles. The replication data in cohort 5 from the GWAS in Central Europeans was provided by Mangold et al.[13], whereas the replication data in cohorts 6 and 7 were from the GWAS of case–parent trios of Asians and European ancestry provided by Beaty et al.[12]. All the controls and cases for each replication cohort were sampled from the same locality and the same population in each study, to assure minimal population stratification effects for each replication.

**DNA extraction.** Approximately 4 ml EDTA anticoagulated venous blood sample was collected from each participant. Genomic DNAs of the cases were extracted from peripheral blood lymphocytes using the standard SDS–proteinase K-phenol/chloroform method. For the controls, DNAs were isolated by standard procedures using Flexi Gene DNA kits (Qiagen) according to the manufacturer's protocol. After quality control, DNAs were diluted to working concentrations of $45–55 \, ng \, \mu l^{-1}$ for genome-wide genotyping and $20–30 \, ng \, \mu l^{-1}$ for the validation studies, respectively.

**Genotyping and quality controls in GWAS.** The discovery-stage genotyping was conducted according to the Infinium HD protocol using the Illumina HumanOmniZhongHua-8 v1.1 BeadChip (Illumina, San Diego, CA, USA) at the Key Laboratory of Dermatology at Anhui Medical University (Ministry of Education), Hefei, Anhui, China. Genotyping was performed as described in the Infinium HD protocol from Illumina[53].

In the GWAS stage, a total of 900,015 SNPs were genotyped in 2,096 cases and 4,051 controls. A standard quality-control criterion was applied to select SNPs and samples for further analysis. SNPs were excluded if they had (i) a call rate <95% in cases or controls; (ii) an MAF of <1% in the population; or (iii) significant deviation from Hardy–Weinberg equilibrium (HWE) in the controls ($P \leq 10^{-4}$). In addition, all the SNPs on the X, Y and mitochondrial chromosomes, as well as the copy number variation-related SNPs and probes, were excluded from statistical analysis. Meanwhile, samples were removed if they (i) had an overall genotyping rate of <98%; (ii) were duplicates or showed familial relationships based on pairwise identity by state using PLINK 1.07 (ref. 54), the sample with higher call rate was left between the related samples (PI_HAT > 0.025); (iii) showed inconsistent genetic gender with epidemiological or clinical data; (iv) and were ancestry outliers or heterozygosity outliers. Samples were assessed for population stratification using the software package EIGENSTRAT[55]. The original script from EIGENSTRAT was modified to extract the principal components for plotting. In total, 63 samples were removed from analysis. After quality control, the genotype data of 803,202 autosomal SNPs in 2,033 cases and 4,051 controls remained for further analysis.

**SNP selection for replication studies.** SNPs were selected for NSCLP replication according to the following steps: (i) we first picked out all the top SNPs with $P < 1.0 \times 10^{-4}$ in the initial stage and excluded the SNPs with ambiguous genotype scatter plots; (ii) then we selected at least one SNP with the lowest $P$-values in each of the novel loci, which defined by using the PLINK option '-indep-pairwise 50 5 0.2'; (iii) in addition, one to four top SNPs were chosen from the previously reported loci; (iv) we also selected SNPs that were located within or close to the susceptibility genes with gene expression profiling evidence for OFC or for syndromes with OFC symptoms. In total, 152 SNPs were selected for the NSCLP replication stage. Furthermore, all the promising SNPs were selected for the NSCLO and NSCPO replications. These SNPs had the lowest $P$-values in NSCLP-meta stage and showed $P < 0.05$, as well as with call rate > 90%, MAF > 0.01 and $P_{HWE} > 10^{-4}$ in NSCLP replication stage; thus, 41 SNPs reached genome-wide significance. The above 41 SNPs were distributed in 12 previously reported NSCL/P associated loci and 14 novel loci. In cohort 5–7 replications,

24 SNPs were selected for replication in 3 GWAS data sets from Central Europeans, Asians and European ancestry groups. Of the 24 SNPs, 20 were picked out from the 14 novel loci and 4 were from 2 newly reported NSCL/P loci (16p13.3 (ref. 15) and 2p24.2 (ref. 16)) and all of them were from the 41 significant SNPs.

**Genotyping and quality control in replication studies.** Genotyping analyses of replications in cohorts 2–4 were conducted by using the Sequenom MassARRAY system, at the Key Laboratory of Dermatology at Anhui Medical University (Ministry of Education), Hefei, Anhui, China. Locus-specific PCR primers (Supplementary Table 15) were designed using MassARRAY Assay Design 3.0 software, following the manufacturer's instructions (Sequenom)[53]. Quality control was performed in each data set separately using PLINK 1.07. In each case–control replications (cohorts 2–4), we excluded SNPs with a call rate <90% in cases or controls, or deviation from HWE proportions ($P \leq 1 \times 10^{-4}$) in the controls.

To evaluate the quality of the genotype data for the validation analyses, 100 randomly selected samples from the GWAS stage were re-genotyped using the Sequenom system. The concordance rate between the genotypes from the Illumina HumanOmniZhongHua-8 v1.1 BeadChip and the Sequenom MassARRAY assay analyses was >99%. The cluster plots from the Illumina and Sequenom analyses were checked to confirm their good quality. After quality control, 146 SNPs were remained for NSCLP replication and 40 SNPs were left for further replications in cohort 3 and 4 analyses, respectively.

**Statistical analyses.** In the GWAS stage, we examined potential genetic relatedness based on pairwise identity by state for all of the successfully genotyped samples using the PLINK 1.07 software. For the duplicated samples and all pairs of first-, second- and third-degree relatives detected, the subject from each pair with the lower call rate was removed from further analysis. All cases and controls were assessed by principal components analysis for population stratification and were confirmed to be of Chinese ancestry. Quantile–quantile plots were constructed and calculations of genomic control values ($\lambda_{GC} = 1.04$ indicated a negligible inflation of the genome-wide statistical results due to population stratification) were performed by using the software R (http://www.r-project.org/) to evaluate the overall significance of genome-wide association results and the potential impact of population stratification, respectively, in the discovery stage.

Association of GWAS and replication analysis were performed using the Cochran–Armitage trend test. Single-marker association analyses were performed to test for disease–SNP associations using logistic regression in each stage. Fixed effects meta-analyses of cohorts 1 and 2 (NSCLP combine) were performed using the Cochran–Mantel–Haenszel test, where $P$-values and heterogeneity index $Q$-values from Cochran's $Q$ statistics were also obtained. Assessment of heterogeneity across studies was carried out by evaluating the $P_{het}$ values from Cochran's $Q$ statistics (Bonferroni-corrected heterogeneity $Q$-values $P_{het}$ of < 0.05 were considered significant)[52,56]. OR values were measured as OR per allele and presented for the minor allele of a SNP, unless otherwise stated. A threshold of $P < 5 \times 10^{-8}$ was adopted to define novel loci with genome-wide significance. The regional association plots for each susceptibility locus were generated in R using information from the HapMap project (CHB and JPT samples). After applying quality control and removing those SNPs with MAF < 1%, HWE < 0.0001 and call rate < 95% from GWAS data set, 2,033 cases and 4,051 controls with 803,202 SNPs were used for the disease variation assessment in the genome-wide level. Furthermore, the samples passed quality control from the discovery and NSCLP replication (3,379 cases and 8,593 controls) with the 41 markers attaining genome-wide significance ($P < 5 \times 10^{-8}$) were used for disease variation estimating of the 26 NSCLP risk loci. The proportion of variance in NSCLP risk was examined via the residual maximum likelihood method in the program genome-wide complex trait analysis and estimated assuming a disease prevalence of 0.001 (1 out of 1,000) and log additive risk[52,57]. All power calculations were performed using the genetic power calculator assuming a disease prevalence of 0.001 and log-additive risk. We carried out conditional analyses to identify additional association signals after accounting for the effects of known and newly discovered susceptibility loci. To investigate more than two association signals per locus, we used a stepwise procedure in which additional SNPs were added to the model according to their conditional $P$-values, as programmed in EMMAX. We estimated the LD metrics $r^2$ and $D'$ using 6,084 individuals from METSIM, who passed genotyping quality control. To replicate associations of the 24 SNPs in different ethnicities, GWAS data from three previously published NSCL/P populations (Central Europeans, Asians and European ancestry groups) were extracted. Replication in the Central European NSCL/P samples was based on a data set published in Mangold et al.[13] SNPs that had not been genotyped in this study were imputed using IMPUTE2 software[7]. Genotype imputation for the case–parent trios described in Beaty et al.[12] was run by the GENEVA Coordinating Center[58], using a worldwide 1,000 Genomes Project reference panel and the IMPUTE2 software in 2012. Imputed genotypes and accompanying marker annotation and quality metrics files are available through the authorized access portion of the dbGaP posting.

**Stratification analyses.** Genotype–phenotype stratification analyses were conducted by using PLINK 1.07 software for the 41 significant associated markers in NSCLP-meta stage. Genotype data were extracted from GWAS and NSCLP

replication stages. Then, we performed stratification analyses on gender and maternal gestational age in NSCLP. $P$-value below $1.22 \times 10^{-3}$ using logistic regression (0.05 out of 41, Bonferroni correction) was considered to be statistically significant.

Heterogeneity analyses among NSCLP, NSCLO and NSCPO were performed by using PLINK 1.07 software based on the 40 significant associated markers in NSCLP meta-stage. Genotype data were extracted from discovery and replication stages of NSCLP, NSCLO and NSCPO (cohorts 1–4). We first divided the cases into three sub-phenotypes NSCLP, NSCLO and NSCPO, then extracted genotype of each case from the above four cohorts and calculated the association between each combination of two sub-phenotypes. $P$-value below $1.25 \times 10^{-3}$ using logistic regression (0.05 out of 40, Bonferroni correction) was considered to be statistically significant.

**Locus annotation and candidate gene prioritization.** To prioritize candidate genes, besides the nearest genes to the index SNPs, the following methods were used to help prioritize potential causal genes in each associated region. All genes located in the same LD block as the index SNPs ($r^2 \geq 0.7$) were selected[52] and annotated for function in molecular, cellular, animal model and tissue/organ levels using several databases, including PubMed (http://www.ncbi.nlm.nih.gov/pubmed/), EMAGE (http://www.emouseatlas.org/emage/home.php), MGI (http://www.informatics.jax.org/), OMIM (http://www.omim.org/), Gene (http://www.ncbi.nlm.nih.gov/gene/), UCSC (http://genome.ucsc.edu/) and Ensembl (http://www.ensembl.org/index.html). The nearest genes on both sides of the index SNP were annotated when no gene was located within the LD block. A total of 135 SNPs at these 26 NSCLP risk loci (all with $r^2 \geq 0.7$ with the SNPs found to be genome-wide significant here) with $MAF > 0.05$ and $P_{HWE} > 1 \times 10^{-4}$ were annotated by using the following methods: regulatory features from ENCODE Consortium/ENCODE/Roadmap Epigenomics Project (http://www.roadmapepigenomics.org/)[59,60].

**Network analysis.** We expanded the global network by including the Human Net protein interaction database[61] and literature-curated interactions from STRING[62,63] to derive an expanded global network based on known protein–protein interactions using the notable genes of the 26 NSCLP associated loci from the present study (Fig. 3).

**GWAS catalogue reviews.** We evaluated all the SNPs within $\pm 500$ kb of the index SNPs (from the 26 loci) and with $P < 5 \times 10^{-8}$ recorded in National Human Genome Research Institute GWAS catalogue (http://www.genome.gov/gwastudies) updated on 20 February 2015. The LD patterns of the index SNPs and the recorded SNPs in GWAS catalogue were inquired using SNAP version 2.2 in Asian (CHB + JPT) and European (CEU) populations using data from the 1,000 Genomes Project Pilot 1.

**Expression studies in the mouse.** Eight- to 14-week-old wild-type Kunming mice were housed in approved specific pathogen-free conditions and mated for 12 h, the presence of a vaginal plug was designated as E0.5. Pregnant mice were randomly divided into four groups and killed at embryonic stages E13.5–E16.5, respectively. Embryos with death or other malformations were ruled out. Normal fetuses were harvested and fixed in 4% paraformaldehyde overnight at 4 °C for IHC. The 4 μm paraffin sections were deparaffinized, rehydrated and subjected to antigen retrieval with high pressure method. A mixture of 30% $H_2O_2$ and methanol (1/9, v/v) was performed to inhibit endogenous peroxidase activity. The rabbit polyclonal antibodies to Fam49a (LS-C167900, LSbio; 1:100 dilution), Rad54b (orb100108, Biorbyt; 1:200 dilution), Rps26 (14909-1-AP, Proteintech; 1:800 dilution), Taf1b (12818-1-AP, Proteintech; 1:600 dilution) and Thap2 (orb186252, Biorbyt; 1:200 dilution) were incubated with the sections at 4 °C overnight, respectively, and were then detected with the Rabbit SP kit (SP9001, Zhongshan Golden Bridge Biotech). The sections were then counterstained with haematoxylin. The results were assessed by an investigator who was blinded to the group allocation. All experimental procedures were carried out in accordance with the Institutional Animal Care and Use Committee of the Laboratory Animal Center of Wuhan University, China. The study was approved by the Ethics Committee, School and Hospital of Stomatology of Wuhan University, China.

**Data availability.** The data that support the findings of this study are available from the corresponding author on request.

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

## Acknowledgements

We thank all the individuals for participating in this project and we acknowledge all oral surgeons at relevant hospitals for their help in the recruitment of subjects, including Hospital of Stomatology of Wuhan University, The Second Charity Hospital of Henan Province, Stomatological Hospital of Nanyang and Stomatological Hospital of Xiangyang, as well as the laboratory staff who contributed to making this work possible. This work was funded by Pre-National Basic Research Program of China (973 Plan; 2012CB722404 to L.D.S. and Z.B.), Top-Notch Young Talents Program of China and Recovery Medical Science Fundation to L.D.S., National Natural Science Foundation of China (81120108010, 81470727, 81300870 and 81571438 to Z.B., M.H. and L.Y.M.), National Key Research and Development Program (2016YFC1000505) and Hubei Province's Outstanding Medical Academic Leader Program to Z.B., and Applied Basic Research Program of Wuhan, China (2014062801011267) to L.Y.M.

## Author contributions

Z.B. and L.D.S. conceived this study. L.D.S., Z.B., M.H. and L.Y.M. provided financial support. Z.B., L.D.S., X.B.Z. and X.J.Z. designed the study. Y.C.F., Y.L.S., Y.C., C.Q.Q., X.Z.F., J.B.H., Z.Y.L., H.S.Y., Z.W.Z., W.J.W., B.L. and Y.Q.Y. conducted sample selection and clinical data management. The following authors from the various collaborating groups undertook data assembly of case/control series in their respective regions and collected data and samples: Y.C.F., Y.L.S., Y.C., C.Q.Q., X.Z.F., H.S.Y. and Y.Q.Y. in Hubei province; J.B.H. and Z.Y.L. in Henan province; W.J.W., Z.W.Z. and B.L. in Anhui province. F.S.Z. and G.C. performed genotyping and sequencing. X.B.Z., X.D.Z., L.D.S., Z.B., M.H., J.P.G., W.J.W. and Y.Q.Y. undertook data manipulation, statistical analysis and bioinformatic interrogations, and data checking. L.Y.M. and X.H.W. conducted animal experiment. T.H.B. and I.R. contributed data from the Baltimore study. E.M. and K.U.L. contributed data from the Bonn-II study. Z.B., L.D.S., M.H., Y.J.S., W.J.W. and Y.Q.Y. contributed to manuscript writing. Z.B., L.D.S., J.J.L., T.H.B., E.M., M.L.M. and K.U.L. helped to revise the manuscript. All authors contributed to the final paper, with Z.B., L.D.S., X.J.Z., Y.Q.Y., X.B.Z., M.H. and J.P.G. playing key roles.

## Additional information

**Competing financial interests:** The authors declare no competing financial interests.

