## [Peer Review File · Nature Communications]

Reviewers' Comments:

Reviewer #1 (Remarks to the Author)

The authors report the largest GWAS for CL/P in a Chinese population. While an earlier GWAS for CL/P in a Chinese population identified one new locus and confirmed four previous loci, with its larger cohorts, this study obtained genome-wide significance with 14 new loci and 12 of 22 loci identified by previous GWAS. They also retested all significant loci for association with their populations stratified by orofacial cleft type (CLO and CPO) and in two non-Chinese populations. They conclude that the new loci "highlighted the important roles of FGF signaling pathway with etiology of NSCLP in a Chinese Han population". Finally, the authors created a series of lists based on searches of publicly available databases in an effort to gain functional evidence to support their genetic data.

Major comments:

The authors performed the largest GWAS to date for CL/P in a Chinese population. The large number of new loci (14) and large fraction of confirmed previous loci (12 of 22) likely reflect the enhanced power of their larger cohort.

This number of total loci afforded the authors the opportunity to screen for genetic pathways involved in orofacial clefting in the Chinese population, an analysis that was not been performed in previous studies. In fact, this unique aspect of the paper is reflected in its title, "...highlighted the important roles of FGF signaling pathway...". However, this conclusion is not supported because the authors did not perform any rigorous and systematic analysis to test for this association. Rather, this conclusion is based on the simple observation that a few of the associated loci contained genes that are directly involved in the FGF-signaling pathway (FGF10, FGFR1, SPRY1) or indirectly involved (PTCH, WNT9B). Moreover, the putative indirect mechanisms are poorly defined.

Does a pattern exist between the 12 previously identified loci that were replicated in this population versus the 10 that were not? For example, did allele frequencies differ significantly? Such patterns could help explain differences in global patterns of disease risk. Similarly, for the 14 new loci, are allele frequencies significantly higher in the Chinese population than in other populations? Or, might these population differences reflect a more subtle biological explanation?

The authors provide lists of functional data about the variants (e-QTLs, location in ENCODE regulatory elements, non-synonymous variants) and genes (expression, mouse phenotypes, biological literature, associated diseases). These lists contain lots of information, but their usefulness is rather limited because they lack an apparent statistical framework and output (as well as a critical assessment by the authors). For example:

- Supp Table 5 lists e-QTLs. These data are derived from PBMCs. How relevant are these data to cell types essential for development of lip and palate? What is expected from a random sampling of variants? Also, the most significant hits in this list at each locus are apparently not to the gene likely to be involved in orofacial development (e.g. IRF6 at 1q32.2), but to genes nearby (TRAF3IP3 at 1q32.2). How should the reader interpret these observations?
- Supp Table 6 lists enhancers/promoters that contain the associated variants. Are there ChIA-PET (or similar) data that directly link these regulatory elements to the candidate genes?
- Supp Table 12 lists the only non-synonymous variant identified. What is the missense change? Like other lists in this report, there is no analysis. For example, is this variant likely to be functional? This could be addressed simply by using web-based analytic tools. In addition, of the

scores of significant variants identified, why is this the only non-synonymous variant? Does this simply reflect a sampling bias in the original chip array? Or, is this expected from a random sampling, or is this biologically significant?

- Finally, can these lists be used to support/refute the authors' conclusion that genes involved in the TGF signaling pathway are over-represented in CL/P loci?

Supp Table 15 contains a list of genes that are associated with orofacial clefting syndromes. Two genes that might be added are RPS26 and VAX1. Mutations in RPS26 cause Diamond-Blackfan Anemia syndrome (DBA10). Patients with DBA can have cleft palate. Mutations in VAX1 cause microphthalmia associated with cleft lip and palate and agenesis of the corpus callosum (MCOPS11). The RPS26 association is interesting because DBAs belong to a class of diseases called ribosomopathies. Included in this group of diseases is Treacher-Collins syndrome - a classic syndrome of cleft palate only, and is caused by mutations in genes that regulate production of rRNA. Another potential link between the genes in this list of CLP loci and ribosomopathies is TAF1B. TAF1B encodes a transcription factor that regulates RNA polymerase 1 expression, which transcribes the rDNA genes. Thus, in addition to FGF-signaling, do the many lists of functional criteria found in the supplementary materials of this paper support the hypothesis that genes in ribosome production/function are involved in orofacial clefts in the Chinese population? One argument against this hypothesis is that neither of these loci were associated with CPO in their Chinese cohort.

Other comments:

The paper is not written well. There are many instances of run-on sentences, and the lack of a helping verb with the verb "locate".

Given the complexity of this study, Supp Figure 1 is an important tool for the reader. However, in its current state, it is difficult to follow. In my understanding of the design, the first arrow labeled "Replication study" is actually the "Discovery study". An arrow for the first "replication study" and its result (confirmation of 64 of 146 SNPs genotyped) is not shown. It might also be helpful to include the cohort names used for each step (e.g. Cohort 1 was used for "Discovery study" and Cohort 2 was used in first "replication study", in addition to the number of cases and controls and geographic origin). After the second replication study, the flow diagram splits into two analyses with "41 SNPs" and "24 SNPs". It might be helpful to reader to list the 26 loci represented by the 41 SNPs and to color code the loci that were identified in this study or confirmed by this study. The final rectangles show the number of significant loci for each analysis. Again, it might be helpful to the reader to include the names of the significant loci in those rectangles. Finally, in this figure, why is the P value for the "Central European" cohort different than the other two non-Chinese cohorts? Is this a typo?

The plots in Supp Figure 4 are an oft used and informative representation of data at individual loci from GWAS. The X-axis is labeled with the chromosome, but would be more informative if it were labeled with the cytogenetic location (e.g. change Chr 1 to Chr 1q32.2). The authors use a 1Mb window for the X axis. While this distance is fine for most loci, there are several loci where the associated variant is located in a gene desert, and the 1Mb window may not be adequate. The best example is the locus at 6p24.3 where the 1Mb window only shows OFCC1, but if expanded to cross the gene desert, the window would show TFAP2A at the centromeric end of this desert. TFAP2A is another strong candidate gene for this locus, as variants in this gene cause and contribute risk for orofacial clefts.

Supp Figure 5 shows immuno-staining for proteins encoded by genes at three loci, RAD54B, RPS26, and FAM49A. The relatively ubiquitous patterns observed are consistent with online databases (MGI and Protein atlas). While it is good to observe expression of candidate gene products in the tissue of interest, the data do not suggest obvious hypotheses for potential

mechanisms.

Reviewer #2 (Remarks to the Author)

This is an important paper that adds very significantly to our understanding of the genetics of CLP. For this reason, I am very enthusiastic about this work. I have some general and specific concerns, though, that are listed below.

General Comments:

- The text is very dense and difficult to read. I wonder if some of this could be improved by moving some of the many p values to tables or not repeating values in the text that are in the tables. Same with gene acronyms and region locations. While important genes should, of course, be mentioned in the text, lists of marker names or chromosomal regions make for very dense reading. While this is critical information, some of it could easily be moved to tables, leaving the text to focus more on the conceptual side of this important work.
- The discussion is pretty terse on what all of this means. I think the volume of work here is one thing but placing it in a broader context is another. Currently, the paper doesn't really do that. I can easily see how this could be done and I believe that this paper has important implications for understanding CLP as a complex trait. These implications are barely touched upon. The fact that, for once, genes like *Wnt9b* pop up that are associated with well studied mouse models, for instance, does not even get a mention. The specific genes that come up in this study are quite interesting in terms of what they suggest about mechanisms of CLP. There are numerous examples of this, actually. This needs to be, at the very least, pointed to! There are well developed hypotheses out there about the developmental mechanisms for CLP. These results are directly relevant to them. Why not state this in the paper!!! The room for this much needed discussion could come from moving a lot of the jargon into the tables. This would also make the paper an easier and more enjoyable read.

Specific comments:

- In supplemental figure 6, it is very difficult to distinguish between the controls and cases. Perhaps use different colours or open versus closed points.
- There are a lot of data in supplementary table 2 and this is important because it gets at the issue of what proportion is replicating and from what kind of underlying distribution. It would be helpful to show these data in graphical form, plotting the effect sizes in the GWAS and replication phases against each other as well as the p values. From those distributions, it should be possible to see if the ones that replicate are simply the tail ends of a continuous distribution or if there is some discontinuity there.
- Similarly, it would be nice to see graphically the overlap between significant SNPs among the populations used here. Some of that is likely sampling or Beavis effect, but some could reflect geographic variation in the genetics of CLP which would be interesting given hypotheses that have been proposed in the past about the developmental basis for geographic variation in CLP. In particular, interaction with the genetics of facial shape would be of interest here and could be discussed depending on what such an analysis shows.

Reviewer #3 (Remarks to the Author)

In the manuscript titled "Genetic studies identified fourteen novel risk loci for NSCLP and highlighted the important roles of FGF signaling pathway with etiology of NSCLP in a Chinese Han population" the authors described a large GWAS that were conducted on the NSCLP as well as NSCLO/NSCPO phenotypes in Han Chinese. This study has focused on a very common birth defect that affects a lot of new-borns. Any valid new findings on this issue would be highly valuable. In general, this study is well structured and ample participants have been analyzed in different steps of their GWAS, which assured the discovery of authentic signals. Indeed, the authors discovered 14 novel loci and confirmed many reported loci associated with NSCLP. The methods and statistical analyses look fine except that the authors have to carefully justify the reason why they used a

preliminary cutoff of $10e-4$ in the discovery cohort? What would happen if they stick to a conventional cutoff of $5 \times 10e-8$? Why is $10e-4$ not too low? Furthermore, it seems to me that their rate of successful replication of the candidate loci in the replication cohort is too high. $10e-4$ is a very very low threshold. In a null condition (no association at all), every 1 out of 10k SNP will achieve this significance just by chance. Given a chip of one million SNPs, we expect to see ~ 100 false positive markers with P values passing this cutoff just by chance. However, the authors chose 146 SNPs in the follow-up cohort and more than 60 SNPs were successfully replicated. This rate of true positive seems a bit too high to me. Other than this issue, the authors should put more words to describe their statistical analyses. For example, in sentence 119, they simply mentioned "A fixed-effects meta-analysis of the combined cohorts..." without give any information how the fixed-effect analysis was conducted. This makes it difficult to evaluate the validity of their analysis. The last major issue is the introduction part. The written English is poor compared to the rest of the manuscript. The authors may ask a native speaker to polish the language. Just to name a few lines that should be carefully revised:

Line 89,

Line 90: "the highest birth prevalence rate of OFC is ...", what does this mean?

Line 92: awkward sentence, need to be re-arranged.

Line 94 - 98: a long sentence with confusing meaning

Minor points:

Line 131: "... using several complementary methods". Please briefly describe what these methods are.

Line 150 - 151, "... and are located at one active ..." awkward writing and confusing meaning.

From line 153 on, the authors claimed that several genes were involved in the FGF pathway, although the following description hardly explained how these genes contribute to FGF pathway. It would be good if the authors can draw a pathway network where the relationship of these genes can be clearly presented.

Line 199: "in near perfect", typo

Line 209, indicate should be indicates

Line 212, " the probably target effector gene", strange sentence with wrong grammar

Line 227 "...were failed..." should be "failed"

Reviewers' comments:

Reviewer #1 (Remarks to the Author):

The authors report the largest GWAS for CL/P in a Chinese population. While an earlier GWAS for CL/P in a Chinese population identified one new locus and confirmed four previous loci, with its larger cohorts, this study obtained genome-wide significance with 14 new loci and 12 of 22 loci identified by previous GWAS. They also retested all significant loci for association with their populations stratified by orofacial cleft type (CLO and CPO) and in two non-Chinese populations. They conclude that the new loci "highlighted the important roles of FGF signaling pathway with etiology of NSCLP in a Chinese Han population". Finally, the authors created a series of lists based on searches of publicly available databases in an effort to gain functional evidence to support their genetic data.

Major comments:

The authors performed the largest GWAS to date for CL/P in a Chinese population. The large number of new loci (14) and large fraction of confirmed previous loci (12 of 22) likely reflect the enhanced power of their larger cohort.

This number of total loci afforded the authors the opportunity to screen for genetic pathways involved in orofacial clefting in the Chinese population, an analysis that was not been performed in previous studies. In fact, this unique aspect of the paper is reflected in its title, "...highlighted the important roles of FGF signaling pathway...". However, this conclusion is not supported because the authors did not perform any rigorous and systematic analysis to test for this association. Rather, this conclusion is based on the simple observation that a few of the associated loci contained genes that are directly involved in the FGF-signaling pathway (FGF10, FGFR1, SPRY1) or indirectly involved (PTCH, WNT9B). Moreover, the putative indirect mechanisms are poorly defined.

Response: Thanks for your comments. Since 26 loci (14 new and 12 previously reported) have been identified in our study, it is possible for us to perform some functional analyses. However, as you indicated, these functional evidences were based on publicly available databases, lacking solid experiments and tests. Hence, we agree that the title of this manuscript over highlighted the importance of FGF-signaling pathway in the pathogenesis of CLP. After serious and thorough discussion within our team, we decided to focus on presenting the genetic implications of these loci for CLP and revised the **Title** as "Genetic studies identified fourteen novel loci for NSCLP and provided new insights into its aetiology" accordingly. In addition, we also diminish the description of FGF signaling pathway in the **Abstract** section. At the same time, several loci

contain candidate genes either related with FGF signaling members or FGF signaling regulators, such as rs10462065 at 5p12 ($P_{\text{meta}} = 1.12 \times 10^{-8}$) adjacent to *FGF10* gene, rs13317 at 8p11.23 ($P_{\text{meta}} = 3.96 \times 10^{-8}$) located in an enhancer within *FGFR1* gene, rs908822 at 4q28.1 ($P_{\text{meta}} = 4.33 \times 10^{-8}$) downstream *SPRY1* gene and rs10512248 at 9q22.32 ($P_{\text{meta}} = 5.10 \times 10^{-10}$) located at the intronic of *PTCH1* gene). And also we performed preliminary functional analysis including GO and network analyses for interpreting the implications of these candidate genes for CL/P. GO analysis show that several candidate genes related with FGF signaling are enriched in fibroblast growth factor receptor binding (FDR = 1.47×10^{-4}), fibroblast growth factor-activated receptor activity (FDR = 2.48×10^{-3}), fibroblast growth factor binding (FDR = 2.53×10^{-2}). The network contains 13 out of the 29 susceptible genes from the present study and shows that several FGF signaling related genes such as *FGFR1* and *FGF10* are connected (**Response letter figure 1**). Based on these evidences, we think it is rational to discuss these FGF signaling related genes as potential candidate ones for CLP.

Response letter figure 1. Functional similarity network of genes in CLP associated loci.

Does a pattern exist between the 12 previously identified loci that were replicated in this population versus the 10 that were not? For example, did allele frequencies differ significantly? Such patterns could help explain differences in global patterns of disease risk. Similarly, for the 14 new loci, are allele frequencies significantly higher in the Chinese population than in other populations? Or, might these population differences reflect a more subtle biological explanation?

Response: Thanks for your comment. We compared the allele frequencies for 9 previously reported SNPs obtained from the 22 reported loci and the 14 newly identified loci and found the existence of the allele frequencies difference for some loci among different populations. The comparisons of allele frequencies for the same markers of these loci between Chinese and other populations could be seen in the following **Response letter table 1** and **2**. For the index SNPs of the 22 reported loci from previous GWASs, most of the allele frequencies can't be obtained, because of their incomplete information in the originally published papers or those SNPs do not exist on our BeadChip. Of the 9 SNPs from 5 out of the 22 loci, 3 of them at 8q24.21 show significantly different between Chinese and other populations based on GWAS data. The allele frequencies of rs987525, rs1530300 and rs7017252 are much lower in Chinese population than in Central European population (**Response letter table 1**). We also compared allele frequency of all the index SNPs (41 SNPs) from the 14 newly identified loci and the 12 reported ones in our study with AFR, AMR, ASN and EUR populations (**Response letter table 2**) based on the 1000 genome project data, which showed that the allele frequencies for some of these SNPs were different among different populations. There is no obvious pattern in terms of allele frequencies for the SNPs from the previously identified and newly identified loci among Chinese and other populations.

Response letter table 1. Comparisons of the allele frequencies for 9 previously reported SNPs obtained from the 22 reported loci among different populations based on GWAS data.

Loci	Index SNPs	Alleles	MAF_cases	MAF_controls	Population	Types
1p22.1	rs481931	A	0.339	/	Asian & European	trios
			0.351	/	European	trios
			0.332	/	Asian	trios
		A/C	0.33	0.37	Chinese Han	case-control
	rs560426	G	0.399	/	Asian & European	trios
			0.471	/	European	trios
		0.342	/	Asian	trios	
	G/A	0.32	0.3	Chinese Han	case-control	
1q32.2	rs861020	A	0.245	/	Asian & European	trios
			0.246	/	European	trios
			0.244	/	Asian	trios
		A/G	0.24	0.19	Chinese Han	case-control
	rs642961	A/G	0.26	/	Norway	trios
			0.24	/	Denmark	trios
		0.25	/	EUROCRAN	trios	
		0.24	/	Europe	trios	
		0.32	/	Philippines	trios	
	A/G	0.24	0.19	Chinese Han	case-control	
8q24.21	rs987525	A/C	0.381(A)	0.202(A)	Central European	case-control
		A/C	0.07	0.07	Chinese Han	case-control
	rs1530300	C/T	0.478(C)	0.301(C)	Central European	case-control
		G/A	0.05	0.04	Chinese Han	case-control
	rs7017252	T/C	0.449(C)	0.393(T)	Central European	case-control
		A/G	0.08	0.05	Chinese Han	case-control

15q13.3	rs1258763	G/A	0.252 (nsCLP_hard+soft)	/	European, Asian, Central European, Mexican, Yemeni	Meta (including trios)
			0.119 (nsCLP_soft)	/	European, Asian, Central European, Mexican, Yemeni	Meta (including trios)
		A/G	0.08	0.07	Chinese Han	case-control
17q22	rs227731	C/A	0.496 (A)	0.416 (C)	Central European	case-control
		C/A	0.36	0.32	Chinese Han	case-control

Response letter table 2. Comparisons of the allele frequencies for all the SNPs from the 14 newly identified and 12 reported loci among different populations based on 1K Genomes Project data.

Loci	Index SNP	BP (hg19)	AFR	AMR	ASN	EUR
1p22.1	rs4147803	94582293	0.40	0.43	0.35	0.37
1p22.1	rs481931	94570016	0.11	0.38	0.36	0.39
1q32.2	rs2064163	210048819	0.51	0.32	0.43	0.16
1q32.2	rs642961	209989270	0.92	0.82	0.81	0.78
1q32.2	rs861020	209977111	0.89	0.82	0.81	0.78
1q32.2	rs9430019	210050794	0.67	0.69	0.73	0.61
2p24.2	rs10172734	16733054	0.62	0.47	0.67	0.30
2p24.2	rs7552	16733928	0.71	0.48	0.70	0.30
2p25.1	rs287980	9971366	0.54	0.80	0.74	0.78
2p25.1	rs287982	9972442	0.56	0.80	0.74	0.78
4p16.2	rs1907989	4818925	0.30	0.43	0.53	0.36
4p16.2	rs34246903	4794195	0.57	0.23	0.38	0.33
4q28.1	rs908822	124906257	0.02	0.02	0.07	0.06
5p12	rs10462065	44068846	0.01	0.09	0.21	0.10
6p24.3	rs9381107	9469238	0.14	0.21	0.35	0.16
8p11.23	rs13317	38269514	0.21	0.28	0.37	0.25
8q21.3	rs1034832	88918331	0.55	0.44	0.38	0.53
8q21.3	rs12543318	88868340	0.80	0.57	0.41	0.66
8q22.1	rs12681366	95401265	0.35	0.49	0.48	0.35
8q22.1	rs957448	95541302	0.29	0.44	0.51	0.32
8q24.21	rs7017252	129950844	0.23	0.23	0.05	0.36
8q24.21	rs7845615	129888794	0.26	0.43	0.78	0.32
9q22.2	rs7871395	92209587	0.30	0.24	0.25	0.21
9q22.32	rs10512248	98259703	0.51	0.21	0.32	0.34
10q25.3	rs6585429	118893231	0.38	0.71	0.41	0.79
12q13.13	rs3741442	53346750	0.11	0.23	0.56	0.02
12q13.2	rs705704	56435412	0.10	0.25	0.25	0.33
12q21.1	rs2304269	72080272	0.0041	0.09	0.46	0.07
12q21.1	rs7967428	72089040	0.12	0.10	0.46	0.06
13q31.1	rs9545308	80639405	0.02	0.17	0.10	0.32
14q22.1	rs7148069	51839645	0.28	0.33	0.18	0.31
14q32.13	rs1243572	95379499	0.83	0.67	0.43	0.79
14q32.13	rs1243573	95379583	0.83	0.67	0.43	0.79
16p13.3	rs17136624	3996282	0.09	0.20	0.22	0.26
16p13.3	rs2283487	3969886	0.64	0.72	0.45	0.67

17p13.1	rs1880646	8929845	0.30	0.34	0.52	0.26
17p13.1	rs2872615	8914693	0.20	0.29	0.45	0.23
17q21.32	rs1838105	45008935	0.55	0.74	0.64	0.65
17q21.32	rs4968247	44988703	0.47	0.65	0.40	0.67
17q22	rs227731	54773238	0.80	0.37	0.34	0.45
20q12	rs6129653	39275603	0.89	0.64	0.66	0.66

NOTE: Ancestry-based groups in this the 1K Genomes Project data: AFR, African; AMR, Americas; EUR, European; ASN, CHB (Han Chinese in Beijing) + JPT (Japanese in Tokyo)

The authors provide lists of functional data about the variants (e-QTLs, location in ENCODE regulatory elements, non-synonymous variants) and genes (expression, mouse phenotypes, biological literature, associated diseases). These lists contain lots of information, but their usefulness is rather limited because they lack an apparent statistical framework and output (as well as a critical assessment by the authors). For example:

- Supp Table 5 lists e-QTLs. These data are derived from PBMCs. How relevant are these data to cell types essential for development of lip and palate? What is expected from a random sampling of variants? Also, the most significant hits in this list at each locus are apparently not to the gene likely to be involved in orofacial development (e.g. IRF6 at 1q32.2), but to genes nearby (TRAF3IP3 at 1q32.2). How should the reader interpret these observations?

Response: Thanks for your comments. Due to most variants identified in GWAS and their proxies are non-coding or intergenic, we expected to identify the causal gene by e-QTL analysis. Since there are no e-QTL public data to cell types of lip and palate, so we only perform the analysis based on the current data derived from PBMCs. Indeed, there are some significant hits in this list at each locus which are apparently not to the gene likely to be involved in orofacial development. It might be difficult to interpret the observations. The analysis of e-QTLs is only one of approaches to confirm the susceptibility locus, which may contain the causal gene of disease. So we decide to delete this supplementary table and the relative description of e-QTL in the revised manuscript.

- Supp Table 6 lists enhancers/promoters that contain the associated variants. Are there ChIA-PET (or similar) data that directly link these regulatory elements to the candidate genes?

Response: Thanks for your comment. We have changed **Supplementary Table 10** to replace **Supplementary Table 6** in revised manuscript. It shows the annotations of 135 SNPs based on current ENCODE data, which only indicated the relationship between regulatory elements and associated variants within each locus. Unfortunately, we can't acquire the ChIA-PET data.

- Supp Table 12 lists the only non-synonymous variant identified. What is the missense change? Like other lists in this report, there is no analysis. For example, is this variant likely to be functional? This could be addressed simply by using web-based analytic tools. In addition, of the scores of significant variants identified, why is this the only non-synonymous variant? Does this

simply reflect a sampling bias in the original chip array? Or, is this expected from a random sampling, or is this biologically significant?

Response: Thanks for your comment. Due to the new associations were located within LD blocks containing multiple genes, we try to investigate whether the index SNP was in high LD with the functional variants in each locus, such as non-synonymous coding variant. If the $r^2 \geq 0.7$ between the SNPs and non-synonymous variants, it might represent the same signal in each locus. After analyzing the entire index SNPs, we only find one SNP rs1838105 was highly associated with non-synonymous variant rs197922 within gene *GOSR2* in our study ($r^2 = 0.98$). And then, we performed the functional analysis of the missense variant (SNP rs197922) using web-based analytic tools. According to PolyPhen-2 analysis, a tool for prediction possible impact of an amino acid substitution on the structure, we found the SNP rs197922 is belong to “benign” with the score of 0.575 (amino acid change: p.Arg67Lys). The *GOSR2* gene encodes a trafficking membrane protein, transports proteins among the medial- and trans-Golgi compartments, which may not directly involve in the mechanisms of orofacial cleft. So we remove the result of the non-synonymous variant in both **Supplementary Table 12** and **main text**.

• Finally, can these lists be used to support/refute the authors' conclusion that genes involved in the TGF (FGF, right?) signaling pathway are over-represented in CL/P loci?

Response: Thanks for your suggestions. In our originally submitted paper, we indeed provide lists of functional data about the variants and genes, which contain lots of information. Some of the information might not be directly support the conclusion that genes involved in the FGF signaling pathway in the mechanism of CL/P. We have deleted these tables, such as e-QTLs and non-synonymous variants analysis, in the revised manuscript. However, the information of gene expression (**Supplementary Table 11**), mouse phenotypes (**Supplementary Table 12**), syndromic OFC (**Supplementary Table 13**) and literature review (**Supplementary Table 14**) might support the susceptibility gene related with FGF signaling pathway involved in the pathogenesis of CL/P.

Supp Table 15 contains a list of genes that are associated with orofacial clefting syndromes. Two genes that might be added are RPS26 and VAX1. Mutations in RPS26 cause Diamond-Blackfan Anemia syndrome (DBA10). Patients with DBA can have cleft palate. Mutations in VAX1 cause microphthalmia associated with cleft lip and palate and agenesis of the corpus callosum (MCOPS11). The RPS26 association is interesting because DBAs belong to a class of diseases called ribosomopathies. Included in this group of diseases is Treacher-Collins syndrome - a classic syndrome of cleft palate only, and is caused by mutations in genes that regulate production of rRNA. Another potential link between the genes in this list of CLP loci and ribosomopathis is TAF1B. TAF1B encodes a transcription factor that regulates RNA polymerase 1 expression, which transcribes the rDNA genes. Thus, in addition to FGF-signaling, do the many lists of functional criteria found in the supplementary materials ofthis paper support the hypothesis that genes in

ribosome production/function are involved in orofacial clefts in the Chinese population? One argument against this hypothesis is that neither of these loci were associated with CPO in their Chinese cohort.

Response: Thanks for your comment. These two genes (*RPS26* and *VAX1*) as well as *TFAP2A* have been added into **Supplementary Table 13** (**Supplementary Table 15** has been changed to **Supplementary Table 13** in the revised paper). Indeed, except FGF-signaling, there are also some genes in ribosome production/function might be involved in orofacial clefts in Chinese population. We have listed these genes in the paper and discussed them in the revised manuscript accordingly (paragraph four of the **Discussion** section). The argument is not against the hypothesis that neither of these loci was associated with CPO in the Chinese cohort. There are several potential reasons as followings: firstly, there might be different susceptibility genes predisposing to CLP and CPO, although there are some shared common genetic factors between CLP and CPO. Secondly, the samples size of replication in CLP or CPO is different, which may lead to insufficient power to replicate these genes in CPO samples.

Other comments:

The paper is not written well. There are many instances of run-on sentences, and the lack of a helping verb with the verb "locate".

Response: Thanks for your comment. We have asked a native speaker to read the revised manuscript and make the language revision.

Given the complexity of this study, Supp Figure 1 is an important tool for the reader. However, in its current state, it is difficult to follow. In my understanding of the design, the first arrow labeled "Replication study" is actually the "Discovery study". An arrow for the first "replication study" and its result (confirmation of 64 of 146 SNPs genotyped) is not shown. It might also be helpful to include the cohort names used for each step (e.g. Cohort 1 was used for "Discovery study" and Cohort 2 was used in first "replication study", in addition to the number of cases and controls and geographic origin). After the second replication study, the flow diagram splits into two analyses with "41 SNPs" and "24 SNPs". It might be helpful to reader to list the 26 loci represented by the 41 SNPs and to color code the loci that were identified in this study or confirmed by this study. The final rectangles show the number of significant loci for each analysis. Again, it might behelpful to the reader to include the names of the significant loci in those rectangles. Finally, in this figure, why is the P value for the "Central European" cohort different than the other two non-Chinese cohorts? Is this a typo?

Response: Thanks for your comment. We have revised the figure according to your suggestion. We used the cohort names for each step, such as Cohort 1 represented for "Discovery study" and Cohort 2 represented for "replication study", in addition to the number of cases and controls and

geographic origin. 152 SNPs were selected in the validation stage, and 146 SNPs passed quality control of which 61 showed consistent directions in their estimated effects on risk with nominal association ($P < 0.05$). We include the cohort names and the number of cases and controls as well as geographic origin for each step. After the meta-analysis of Cohort 1 and 2, the flow diagram was split into two analyses, and we list the 26 loci represented by the 41 SNPs. The red color represented the loci that were newly identified in this study; the blue one represented the 12 confirmed by this study. The final rectangles show the number of significant loci for each analysis. The significant loci and their names for each analysis were showed. It is a typo for the P value for the "Central European" cohort different from the other two non-Chinese cohorts and we have revised it in the Figure.

The plots in Supp Figure 4 are an oft used and informative representation of data at individual loci from GWAS. The X-axis is labeled with the chromosome, but would be more informative if it were labeled with the cytogenetic location (e.g. change Chr 1 to Chr 1q32.2). The authors use a 1Mb window for the X axis. While this distance is fine for most loci, there are several loci where the associated variant is located in a gene desert, and the 1Mb window may not be adequate. The best example is the locus at 6p24.3 where the 1Mb window only shows OFCC1, but if expanded to cross the gene desert, the window would show TFAP2A at the centromeric end of this desert. TFAP2A is another strong candidate gene for this locus, as variants in this gene cause and contribute risk for orofacial clefts.

Response: Thanks for your comment. We have changed the X-axis labeled with the cytogenetic location (e.g. change Chr 1 to Chr 1q32.2). If the locus where the associated variant is located in a gene desert, we have revised the window from 1Mb to 2Mb in order to find the candidate gene for the locus (4q28.1 and 6p24.3). We have revised the figures in the **Supplementary Figure 5**(**Supplementary Figure 4** has been changed to **Supplementary Figure 5** in the revised paper). It might be helpful to identify the susceptibility genes that contribute to risk for orofacial clefts.

Supp Figure 5 shows immuno-staining for proteins encoded by genes at three loci, RAD54B, RPS26, and FAM49A. The relatively ubiquitous patterns observed are consistent with online databases (MGI and Protein atlas). While it is good to observe expression of candidate gene products in the tissue of interest, the data do not suggest obvious hypotheses for potential mechanisms.

Response: Thanks for your comment. On the online databases (MGI and Protein atlas), RAD54B does not have any expression data, RPS26 ubiquitously expressed on the whole mouse embryo and expression of FAM49A was observed at the nervous system. In the current stage, we only perform the immunohistochemistry (IHC) analysis on mouse embryos and the preliminary aim was to observe the expression of candidate gene products in the tissues of interest. It shows that

positive IHC staining of the three interested genes (*Rad54b*, *Rps26* and *Fam49a*) in the palatal mesenchymal cells and epithelium cells during the critical period of lip and palatal development, which indicates that these genes indeed play roles in development of the lip and palatal. Although the data does not suggest obvious hypotheses for potential mechanisms, the further study is needed to explore the potential mechanisms of these genes involved the pathogenesis of orofacial clefts.

Reviewer #2 (Remarks to the Author):

This is an important paper that adds very significantly to our understanding of the genetics of CLP. For this reason, I am very enthusiastic about this work. I have some general and specific concerns, though, that are listed below.

General Comments:

The text is very dense and difficult to read. I wonder if some of this could be improved by moving some of the many p values to tables or not repeating values in the text that are in the tables. Same with gene acronyms and region locations. While important genes should, of course, be mentioned in the text, lists of marker names or chromosomal regions make for very dense reading. While this is critical information, some of it could easily be moved to tables, leaving the text to focus more on the conceptual side of this important work.

Response: Thanks for your comment. We have removed the repeated *P* values, gene acronyms and region locations and mentioned the important genes in the revised text according to your comments. The revised manuscript focuses more on the conceptual side of this work.

The discussion is pretty terse on what all of this means. I think the volume of work here is one thing but placing it in a broader context is another. Currently, the paper doesn't really do that. I can easily see how this could be done and I believe that this paper has important implications for understanding CLP as a complex trait. These implications are barely touched upon. The fact that, for once, genes like *Wnt9b* pop up that are associated with well studied mouse models, for instance, does not even get a mention. The specific genes that come up in this study are quite interesting in terms of what they suggest about mechanisms of CLP. There are numerous examples of this, actually. This needs to be, at the very least, pointed to! There are well developed hypotheses out there about the developmental mechanisms for CLP. These results are directly relevant to them. Why not state this in the paper!!! The room for this much needed discussion could come from moving a lot of the jargon into the tables. This would also make the paper an easier and more enjoyable read.

Response: Thanks for your comment. As you suggested, we have modified and enriched the part of discussion in this paper. Particularly, we have discussed some special genes which might be involved in the mechanism of CLP. Please see details in **Discussion** part in revised manuscript (paragraph one and two in **Discussion** section).

Specific comments:

In supplemental figure 6, it is very difficult to distinguish between the controls and cases. Perhaps use different colors or open versus closed points.

Response: Thanks for your comment. We have used different colors to distinguish the controls (red) and cases (green) as you suggested in **Supplementary Figure 3** (**Supplementary Figure 6** has been changed to **Supplementary Figure 3** in the revised paper).

There are a lot of data in supplementary table 2 and this is important because it gets at the issue of what proportion is replicating and from what kind of underlying distribution. It would be helpful to show these data in graphical form, plotting the effect sizes in the GWAS and replication phases against each other as well as the p values. From those distributions, it should be possible to see if the ones that replicate are simply the tail ends of a continuous distribution or if there is some discontinuity there.

Response: Thanks for your comment. We have drawn the **Response letter figure 2** based on the data in **Supplementary table 2** to show the graphical form, and plotted the effect sizes in the GWAS and replication phases through comparing odds ratios for variants in discovery and replication phases. For each SNP, odds ratios (on a log scale) were estimated within different stage. The color of each dot denotes the association P value for each SNP in CLP. The red line denotes the best-fitting least-squares regression line, weighted by the inverse of the variance of the $\log(\text{ORs})$ in CLP. Significance and goodness-of-fit are shown in blue. From those distributions, it can be seen that the successfully replicated SNPs distribute at the tail ends of the fitted line. The figure is really concise, but it can't cover all the data in supplementary table 2. So we keep **supplementary table 2** in our paper, and show the figure here.

Response letter figure 2. The plot of effect sizes in the GWAS and replication phases with P values. (Yellow: the P value of SNPs does not achieve the GWAS significance; Blue: the P value of SNPs achieves the GWAS significance ($P < 5.00 \times 10^{-8}$))

Similarly, it would be nice to see graphically the overlap between significant SNPs among the populations used here. Some of that is likely sampling or Beavis effect, but some could reflect geographic variation in the genetics of CLP which would be interesting given hypotheses that have been proposed in the past about the developmental basis for geographic variation in CLP. In particular, interaction with the genetics of facial shape would be of interest here and could be discussed depending on what such an analysis shows.

Response: Thanks for your comment. We draw the following figure to show graphically the overlap between significant SNPs among different populations used here. It is clear to see the shared and specific common susceptibility loci among different populations. Indeed, there are some loci, such as 2p25.1, 4q28.1, 5p12 that are unique in Chinese or 15q22.2 is unique in European population and some of them including 1q32.2, 8q24.21, 10q25.3, 13q31.1 and 17q22 are shared between different populations (**Response letter figure 3**). It might suggest the existence of geographic variation in the genetics of CL/P. So far, two GWASs conducted by two groups have reported 7 loci associated with facial shape in two different ethnicities (*PRDM16*, *PAX3*, *TP63*, *C5orf50*, and *COL17A1* in European population and *SCHIP1*, *PDE8A* in African population) and all these loci/genes were not identified in our CLP GWAS. Additionally, one of the groups tested associations between facial phenotypes and 11 SNPs originally discovered in previous GWAS on NSCL/P. Five SNPs at 4 candidate NSCL/P loci (2p21, 8q24, 13q31, and

17q22) were significantly associated with facial phenotypes. These results suggested that genetic variants associated with NSCL/P may also influence facial shape. However, we don't have the data for facial phenotypes in our patients with orofacial cleft, thus, such an analysis of interaction with the genetics of facial shape could not be performed at the present time.

Response letter figure 3. The susceptibility loci identified in different populations. Different colors represent the shared common loci or unique loci among the three groups.

Reviewer #3 (Remarks to the Author):

In the manuscript titled "Genetic studies identified fourteen novel risk loci for NSCLP and highlighted the important roles of FGF signaling pathway with etiology of NSCLP in a Chinese Han population" the authors described a large GWAS that were conducted on the NSCLP as well as NSCLO/NSCPO phenotypes in Han Chinese. This study has focused on a very common birth defect that affects a lot of new-borns. Any valid new findings on this issue would be highly valuable. In general, this study is well structured and ample participants have been analyzed in different steps of their GWAS, which assured the discovery of authentic signals. Indeed, the authors discovered 14 novel loci and confirmed many reported loci associated with NSCLP. The methods and statistical analyses look fine except that the authors have to carefully justify the reason why they used a preliminary cutoff of $10e-4$ in the discovery cohort? What would happen if

they stick to a conventional cutoff of 5×10^{-8} ? Why is 10^{-4} not too low? Furthermore, it seems to me that their rate of successful replication of the candidate loci in the replication cohort is too high. 10^{-4} is a very very low threshold. In a null condition (no association at all), every 1 out of 10k SNP will achieve this significance just by chance. Given a chip of one million SNPs, we expect to see ~ 100 false positive markers with P values passing this cutoff just by chance. However, the authors chose 146 SNPs in the follow-up cohort and more than 60 SNPs were successfully replicated. This rate of true positive seems a bit too high to me. Other than this issue, the authors should put more words to describe their statistical analyses. For example, in sentence 119, they simply mentioned "A fixed-effects meta-analysis of the combined cohorts..." without give any information how the fixed-effect analysis was conducted. This makes it difficult to evaluate the validity of their analysis. The last major issue is the introduction part.

Response: Thanks for your comments. The reason why we used a preliminary cutoff of 10^{-4} in the discovery cohort is to have more chance to find more new loci for NSCLP. If we stick to a conventional cutoff of 5×10^{-8} , there are only few SNPs available to be replicated. It seems that the rate of successful replication is too high, however, we selected the 152 privileged SNPs (146 of the 152 SNPs passed quality control) from more than 1000 top SNPs ($P < 10^{-4}$) based on the discover stage after stringent quality control including very good genotype cluster, moderate allele frequencies and conditional analysis aimed to distinguish independent signals. Furthermore, the high rate of successful replication might also be due to the pure phenotype of the cases and controls and relative large sample size both in the discovery and the replication stages. We have described statistical analyses, such as fixed-effects meta-analysis of the combined cohorts as others, the GWAS of leprosy that the co-operated subjects of our team (Nat Genet. 2015;47(3):267-71.). Fixed-effects meta-analysis of the two independent studies (Cohort 1 and 2) was performed using the inverse variance method implemented in META version 1.3.2, where P values from Cochran's Q statistics were also obtained. The heterogeneity among studies was quantitatively assessed using Cochran's Q test (Bonferroni-corrected heterogeneity P values of < 0.05 were considered significant). If the multiple correction of the P value of Cochran's Q test is > 0.05 , then fixed-effects model was used. Otherwise, the random-effects model was used if the multiple correction of the P value of Cochran's Q test is ≤ 0.05 . And also, we have revised the introduction part accordingly.

The written English is poor compared to the rest of the manuscript. The authors may ask a native speaker to polish the language. Just to name a few lines that should be carefully revised:

Response: Thanks for your comment. We have asked a native speaker to read the revised manuscript and make the language revision.

Line 89,

Line 90: "the highest birth prevalence rate of OFC is ...", what does this mean?

Response: Thanks for your comment. This sentence has been revised as "In general, the highest birth prevalence rates of OFC are reported in Asia (especially China and Japan), often as high as 1 in 500 and affecting more than 2.6 million people in China." (Line 87-89)

Line 92: awkward sentence, need to be re-arranged.

Response: Thanks for your comment. The sentence has been re-arranged as "According to whether the patients have other malformations or anomalies, OFC can be divided into syndromic and non-syndromic forms. And approximately 70% of cleft lip with or without cleft palate (CL/P) cases and 50% of cleft palate only (CPO) cases occur as isolated entities with no other abnormal phenotypes are considered to be non-syndromic (referred to as NSOFC)." (Line 90-94)

Line 94 - 98: a long sentence with confusing meaning

Response: Thanks for your comment. We have revised the long sentence to short sentence in order to make it clear, as "NSOFC is further classified into NSCLP, non-syndromic cleft lip only (NSCLO) and non-syndromic cleft palate only (NSCPO) based on the anatomical morphology. Because they share common epidemiological patterns and occur during the same embryological period, NSCLP and NSCLO are often grouped together as non-syndromic cleft lip with or without cleft palate (NSCL/P), differing only in severity. However, there is some evidence showing that NSCLP and NSCLO might harbour different genetic aetiologies." (Line 94-100)

Minor points:

Line 131: "... using several complementary methods". Please briefly describe what these methods are.

Response: Thanks for your comment. We have briefly described these methods accordingly in revised paper (see **Biological implications analyses for the 26 NSCLP loci** in **Results** section and **Locus annotation and candidate gene prioritization** in **Methods** section). (Line 198-221 and Line 532-548)

Line 150 - 151, "... and are located at one active ..." awkward writing and confusing meaning.

Response: Thanks for your comment. We have revised it as "At 12q21.1, the signals are near the *TMEM19* gene, involving the SNPs rs2304269 and rs7967428, which are in strong linkage disequilibrium (LD) with each another ($r^2 = 0.98$). Rs2304269 and rs7967428 are respectively

located at one active promoter and five strong enhancers in epidermal keratinocytes according to ENCODE data.”. (Line 269-273)

From line 153 on, the authors claimed that several genes were involved in the FGF pathway, although the following description hardly explained how these genes contribute to FGF pathway. It would be good if the authors can draw a pathway network where the relationship of these genes can be clearly presented.

Response: Thanks for your comment. We draw a pathway network where the relationship of these genes in this study. The functional similarity network is constructed for the genes reported in the present study (**Response letter figure 1**). The network contains 13 out of the 29 susceptible genes from the present study, such as *FGFR1* and *FGF10*, which may indicate the FGF signaling associated with orofacial clefting.

Response letter figure 1. Functional similarity network of genes in CLP associated loci.

Line 199:" in near perfect", typo

Response: Thanks for your comment. The typo had been revised. (Line 296)

Line 209, indicate should be indicates

Response: Thanks for your comment. The error had been revised. (Line 303)

Line 212, " the probably target effector gene", strange sentence with wrong grammar

Response: Thanks for your comment. We have revised the typographical error “the probably target effector gene” to “the probably target effect gene” accordingly. (Line 305-306)

Line 227 "...were failed..." should be "failed"

Response: Thanks for your comment. We have rearranged the sentence and cut part of the details (including the part contained the word “failed”) because of reorganizing the structure of the article.

Reviewers' Comments:

Reviewer #1 (Remarks to the Author)

Review of revised manuscript

Major comments:

None.

Minor comments:

Thank you for revising Suppl Figure 1 (study design). This is a large figure, but it might be worth adding to the body of the paper.

Typo: The Manhattan Plot is listed as "Supplementary figure 2" in the RESULTS. However, it is now Figure 1 in revised manuscript. Please confirm names in text and in supplementary data.

It is surprising that 8q24 locus was associated with CPO in this study. Previous studies (Birnbaum et al., 2009; Beaty et al., 2010) showed that this locus maybe the single most important locus for CLP in European populations. However, it is not associated with CLP in Asian populations, where the MAF is very low. Also, this locus was not associated with CPO in a European cohort, though the MAF is much higher in that population (Mangold et al., 2010). These are two striking population differences at this locus.

Thank you for comparing the MAF between populations for SNPs that were associated in the Chinese populations but not in previous GWA studies. The data in Tables 1 and 2 of response to reviewers show that all SNPs are just as common in European populations (except the 12q21.1 locus). Thus, the absence of a genome-wide signal in the previous GWA studies is not due simply to differences in allele frequency between populations, as noted by the authors. The general negative result is addressed appropriately in the text. However, what about the counter hypothesis - is there a signal for the novel associated SNPs from this study in the previous GWAS (although such a signal did not reach genome-wide significance)? The testing of this hypothesis may be beyond the scope of this study. However, since the corresponding authors of the previous GWAS are co-authors on this paper, the data should be accessible.

Thank you for performing the GO analysis on the associated loci. A couple comments: The chart used ABCA4 as the gene at 1p21. Although the associated SNP is located in ABCA4, multiple studies suggest that ARHGAP29, the adjacent gene, is the best candidate gene at that locus. The GO analysis also included multiple genes from a single locus, e.g. DCAF4L2 and MMP16 at 8q21.3. Should only one gene should be chosen per locus (unless there is some evidence to suggest multiple independent signals)? Also, if this figure were to be added to the manuscript, then the legend should include more detail about symbols and color coding.

The following sentence is too long. "To further assess the possible
340 independence among these various birth defects/diseases/traits within these particular
341 SNPs, we examined LD patterns between these SNPs in Asian, African and European
342 populations using data from the 1000 Genomes Project and identified three
343 susceptibility loci shared by NSCLP and schizophrenia at 8p11.23, by asthma,
344 polycystic ovary syndrome, rheumatoid arthritis, vitiligo, type 1 diabetes
345 autoantibodies, alopecia areata and NSCLP at 12q13.2, and between height and
346 NSCLP at 9q22.32, where SNPs reported to be associated with NSCLP and other
347 diseases/traits were in strong LD ($r^2 \geq 0.7$), which showed significant and
348 non-independent association of the risk of NSCLP and other diseases/traits."

Reviewer #2 (Remarks to the Author)

This paper is much improved and represents a substantial and very interesting contribution to the field.

The response to reviewers is quite thorough and convincing. However, as a general comment, the additional analyses requested are not solely for the benefit of the reviewers but are intended to improve the paper. I would like to see response letter figures 1 and 3, in particular, incorporated into the paper. The first of these figures I could even see in the main paper and not in supplementary as it is quite interesting.

Suppl Fig2: "Evidences" should not be plural in the figure legend (or elsewhere in the paper).

The title should be written in the present tense rather than past tense.

Reviewer #3 (Remarks to the Author)

The authors have addressed all the questions and comments properly. I do not have further questions. I recommend this manuscript to be published in the current form

REVIEWERS' COMMENTS:

Reviewer #1 (Remarks to the Author):

Review of revised manuscript

Major comments:

None.

Minor comments:

Thank you for revising Suppl Figure 1 (study design). This is a large figure, but it might be worth adding to the body of the paper.

Response: Thanks for your comments. We have added the figure to the body of our paper according to your suggestion. The Supplementary Figure 1 (study design) is changed to "**Figure 1**" in the revised manuscript now.

Typo: The Manhattan Plot is listed as "Supplementary figure 2" in the RESULTS. However, it is now Figure 1 in revised manuscript. Please confirm names in text and in supplementary data.

Response: Thanks for your keen observation and comments. We neglected to mark "**Figure 1**" in main text of the previous revised manuscript. The Manhattan plot in previous revised manuscript (previous **Figure 1**) is a simplified graphic, to highlight the distributions and *P* values of the 14 novel and 12 previously reported loci of NSCLP (It is "**Figure 2**" now). While, "**Supplementary figure 2**" in the RESULTS is a detailed Manhattan plot to show the *P* values of all the 803,202 SNPs in discovery GWAS stage (It is "**Supplementary Figure 1**" now).

It is surprising that 8q24 locus was associated with CPO in this study. Previous studies (Birbaum et al., 2009; Beaty et al., 2010) showed that this locus may be the single most important locus for CLP in European populations. However, it is not associated with CLP in Asian populations, where the MAF is very low. Also, this locus was not associated with CPO in a European cohort, though the MAF is much higher in that population (Mangold et al., 2010). These are two striking population differences at this locus.

Response: It is really interesting that 8q24 is associated with CPO in our Chinese cohort at replication stage. However, this locus was not associated with CPO in a European cohort. Significant associated markers (rs7845615 and rs7017252) were found in CLP GWAS in our study (**Supplementary Table 4**). The two markers were then chosen to be replicated in CPO replication stage, and one of them (rs7017252) showed significant association (**Supplementary Table 7**). In addition, MAF of rs7017252 is also very low in Chinese Han (MAF = 0.07 in CPO cases), and rs7017252 locates very near to rs987525 (approximately 5kb) which is strongly associated with CLP in Europeans. Thank you for pointing out this interesting and important issue. Further study is warranted to investigate potential reasons, especially by performing GWAS in Chinese CPO samples.

Thank you for comparing the MAF between populations for SNPs that were associated in the Chinese populations but not in previous GWA studies. The data in Tables 1 and 2 of response to

reviewers show that all SNPs are just as common in European populations (except the 12q21.1 locus). Thus, the absence of a genome-wide signal in the previous GWA studies is not due simply to differences in allele frequency between populations, as noted by the authors. The general negative result is addressed appropriately in the text. However, what about the counter hypothesis - is there a signal for the novel associated SNPs from this study in the previous GWAS (although such a signal did not reach genome-wide significance)? The testing of this hypothesis may be beyond the scope of this study. However, since the corresponding authors of the previous GWAS are co-authors on this paper, the data should be accessible.

Response: Thanks for your comments. Because the Beadchips used in our study and the previous GWA studies are different, most of the original data of the novel associated SNPs from our study can not be directly acquired in the previous GWAS database from our co-authors. Thus they imputed the corresponding results of all the novel associated SNPs, which are listed in **Table 1** and **Supplementary Table 9**. Since we could not directly compare the genotype data for these SNPs within novel signals between our study and previous study, it is hard to tell the truth. At least, we did not find the association signal in our study for the same index SNP at the previously reported loci. Further study is needed to explore this issue.

Thank you for performing the GO analysis on the associated loci. A couple comments: The chart used ABCA4 as the gene at 1p21. Although the associated SNP is located in ABCA4, multiple studies suggest that ARHGAP29, the adjacent gene, is the best candidate gene at that locus. The GO analysis also included multiple genes from a single locus, e.g. DCAF4L2 and MMP16 at 8q21.3. Should only one gene should be chosen per locus (unless there is some evidence to suggest multiple independent signals)? Also, if this figure were to be added to the manuscript, then the legend should include more detail about symbols and color coding.

Response: Thanks for your comments. We choose *ARHGAP29* instead of *ABCA4* as candidate gene at 1p21 as you suggested. At 8q21.3, only *MMP16* is left after consideration. While at 6p24.3 (*OFCC1 / TFAP2A*) and 14q32.13 (*GSC / DICER1*), two genes from each locus are remained for their potential functional effect to CLP. We then reorganized the network figure and added it to our revised manuscript as **Figure 3**.

The following sentence is too long. “To further assess the possible
340 independence among these various birth defects/diseases/traits within these particular
341 SNPs, we examined LD patterns between these SNPs in Asian, African and European
342 populations using data from the 1000 Genomes Project and identified three
343 susceptibility loci shared by NSCLP and schizophrenia at 8p11.23, by asthma,
344 polycystic ovary syndrome, rheumatoid arthritis, vitiligo, type 1 diabetes
345 autoantibodies, alopecia areata and NSCLP at 12q13.2, and between height and
346 NSCLP at 9q22.32, where SNPs reported to be associated with NSCLP and other
347 diseases/traits were in strong LD ($r^2 \geq 0.7$), which showed significant and
348 non-independent association of the risk of NSCLP and other diseases/traits.”

Response: Thanks for your comments. We have revised the long sentence as “To further assess the possible independence among these various birth defects/diseases/traits within these particular SNPs, we examined LD patterns between these SNPs in Asian, African and European populations using data from the 1000 Genomes Project. As a result, three susceptibility loci were identified to

be shared by NSCLP and other diseases/traits, including schizophrenia at 8p11.23, asthma, polycystic ovary syndrome, rheumatoid arthritis, vitiligo, type 1 diabetes autoantibodies and alopecia areata at 12q13.2, and height at 9q22.32. The SNPs reported to be associated with NSCLP and these diseases/traits were in strong LD ($r^2 \geq 0.7$), which showed significant and non-independent association of the risk of NSCLP and other diseases/traits.”

Reviewer #2 (Remarks to the Author):

This paper is much improved and represents a substantial and very interesting contribution to the field.

Response: Thanks for your recognition.

The response to reviewers is quite thorough and convincing. However, as a general comment, the additional analyses requested are not solely for the benefit of the reviewers but are intended to improve the paper. I would like to see response letter figures 1 and 3, in particular, incorporated into the paper. The first of these figures I could even see in the main paper and not in supplementary as it is quite interesting.

Response: Thanks for your comments. We have added **response letter figures 1** (It is “**Figure 3**” now) into the main paper and **response letter figures 3** (It is “**Supplementary Figure 4**” now) into the supplementary information.

Suppl Fig2: “Evidences” should not be plural in the figure legend (or elsewhere in the paper).

Response: Thanks for your comments. The error had been revised.

The title should be written in the present tense rather than past tense.

Response: Thanks for your comments. We have revised the **Title** as “Genome-wide analyses of non-syndromic cleft lip with palate identify fourteen novel loci and genetic heterogeneity”.

Reviewer #3 (Remarks to the Author):

The authors have addressed all the questions and comments properly. I do not have further questions. I recommend this manuscript to be published in the current form.

Response: Thanks for your recognition.